# An Overview of Analytical Methods to Determine Pharmaceutical Active Compounds in Aquatic Organisms

**DOI:** 10.3390/molecules27217569

**Published:** 2022-11-04

**Authors:** María del Carmen Gómez-Regalado, Laura Martín-Pozo, Julia Martín, Juan Luis Santos, Irene Aparicio, Esteban Alonso, Alberto Zafra-Gómez

**Affiliations:** 1Department of Analytical Chemistry, Sciences Faculty, University of Granada, E-18071 Granada, Spain; 2Departamento de Química Analítica, Escuela Politécnica Superior, Universidad de Sevilla, C/Virgen de África 7, E-41011 Seville, Spain; 3Instituto de Investigación Biosanitaria ibs, E-18016 Granada, Spain

**Keywords:** pharmaceuticals, contamination, analytical methods, aquatic organisms, trophic chain

## Abstract

There is increasing scientific evidence that some pharmaceuticals are present in the marine ecosystems at concentrations that may cause adverse effects on the organisms that inhabit them. At present, there is still very little scientific literature on the (bio)accumulation of these compounds in different species, let alone on the relationship between the presence of these compounds and the adverse effects they produce. However, attempts have been made to optimize and validate analytical methods for the determination of residues of pharmaceuticals in marine biota by studying the stages of sample treatment, sample clean-up and subsequent analysis. The proposed bibliographic review includes a summary of the most commonly techniques, and its analytical features, proposed to determine pharmaceutical compounds in aquatic organisms at different levels of the trophic chain in the last 10 years.

## 1. Introduction

Pollution is one of the biggest environmental challenges worldwide. Like climate change or the depletion of water supplies, pollution threatens the stability of the earth’s support systems and is a growing concern for human health [1]. Ocean pollution is a very important, but under-recognised, component of global pollution [2]. Seawater covers 97% of surface waters and is considered one of the most abundant resources on our planet [1]. The unsustainable use of marine waters and resources by humans has altered the structure of marine ecosystems, relating to the phenomenon of eutrophication, loss of diversity or the presence of polluting chemicals [3].

Human activities have introduced a large number of contaminants of emerging concern (CECs) into the environment [4]. CECs include a wide variety of compounds such as disinfection by-products, natural toxins, flame retardants, personal care products or pharmaceutical active compounds (PhACs) [5]. Nowadays, an increasing number of people and animals are in need of health care, which means that the number and amount of PhACs consumed, and consequently excreted, is very high [6,7,8]. Approximately 3000 compounds are used as pharmaceuticals, with an annual production exceeding hundreds of tonnes [7]. It is well known that the wastewater treatment plants (WWTPs) are often unable to remove them completely, allowing their release into the environment [9,10]. In the case of PhACs, due to their constant release into the seas, even those that can undergo degradation may behave as pseudopersistent contaminants [11]. This continued exposure may present unexpected risks in the organisms that inhabit them such as reproductive disorders, survival of susceptible species, growth rate or development of bacterial resistance and endocrine disruption, among others [8,12,13].

The European Union has developed several laws for the monitoring and protection of the seas and their ecosystem. The Water Framework [14] and the Marine Strategy Framework Directive [15] are based on the maintenance as well as the protection and restoration of the marine environment. In addition, the European Commission has drawn up a first list for the monitoring of CECs in 2015, and then it was updated in 2018, 2020 and 2022. The decision 2022/1307/EC [16], includes some PhACs such as the antibiotics sulfamethoxazole and trimethoprim, or the antidepressant venlafaxine and its main metabolite, O-desmethylvenlafaxine, with a maximum permitted detection limit of 100 ng g^−1^ for the antibiotics and 6 ng g^−1^ for the others. Although quantitative analysis of PhACs in aquatic ecosystems is limited, as dilution makes detection difficult, the use of bioindicator species is valuable in assessing system contamination, since they are able to reflect bioavailability in a variability of concentrations in both water and sediment [11].

Due to the evidence of the presence of these active compounds in the environment and the concern that it raises, as well as the published EU directives, there is a need for the development of analytical methods with the appropriate characteristics to determine these PhACs in biomarkers. Furthermore, taking into account that the PhAC’s consumption depends on factors such as seasonal diseases, the health system and prescribing practices or the economic level of the population, the methodology developed must take into account local needs [10]. The present work deals with a comprehensive overview of the recent methods proposed for the determination of several groups of PhACs in aquatic organisms belonging to different levels of the trophic chain, emphasizing the sample treatment and contrasting the analytical results obtained. For that, we have focused mainly on methodological studies that include analytical quality parameters and relay on liquid chromatography (LC), the most useful separation technique for the multiresidue determination of PhACs [17]. Huerta et al. [18] already reviewed the state of the art of the analysis of PhACs in aquatic biota up to 2012. Thus, the present review provides an update on the current analytical methods since 2012 onwards.

## 2. Multi-Level Biological Groups as Biomarkers of Exposure

Biomarkers are defined as suborganic changes that occur at the cellular, physiological or molecular level, measurable in cells or tissues of an organism, which may be indicative of exposure [19]. To be a useful bioindicator, an organism must have certain characteristics such as a wide geographical distribution, long life duration, being easy to capture, a feeding mode that allows the accumulation of contaminants present in the environment (e.g., filtration) or the ability to accumulate and tolerate high concentrations of organic and inorganic contaminants in their tissues [20,21]. The use of sentinel species to monitor environmental pollution allows knowledge of the bioavailability of pollutants in the environment over prolonged periods of time [22]. In addition, information on the concentration of pollutants in different organisms is quite useful for considering toxicological and public health aspects of pollution in natural systems [23]. Among the distinct species used as bioindicators, fish and bivalves, particularly mussels, stand out, as the latter are present on coasts all over the world, are easy to capture and are filter-feeders [24,25]. However, it is necessary to study pollution in species other than mussels to assess trophic transfer in aquatic ecosystems. Figure 1 displays the number of studies devoted to the analysis of PhACs for each group of marine organisms according to the literature consulted in scientific databases. It shows that fish have been by far the most investigated in this field. This section summarises the use of some species belonging to the diverse levels of the trophic chain as bioindicators of pollution.

### 2.1. Phytoplankton

Phytoplankton is the group of organisms that form part of the exclusively plant-based plankton. They underlie productivity in aquatic environments and are widely used as biomarkers. Among the different species, pigments and fatty acids are mainly used in the study of pollution [3]. Primary aquatic production is carried out by phytoplanktons, which absorb pollutants from the surrounding water and incorporate large quantities into their cellular compartments. In the case, for example, of arsenic, it has been shown that phytoplankton can excrete it after metabolization into the environment, transferring it to higher trophic levels [26,27,28]. Yan et al. [29] studied the bioaccumulation of antibiotics and analgesics in cyanobacteria as target organisms.

### 2.2. Zooplankton

Zooplankton is the fraction of exclusively animal organisms that are part of the plankton. They are very sensitive indicators of the ecological state of an aquatic system since they are able to respond rapidly to environmental changes with modifications in their composition and structure [30]. Zooplankton has an ecologically important role in marine ecosystems being the primary consumer of the food chain. Furthermore, depending on their life stage and the availability of prey, their feeding behaviour varies, being able to combine the selection with chemoreceptors and mechanoreceptors [31]. The same authors mentioned in the previous section also investigated the bioaccumulation of PhACs in several zooplankton species including Daphnia magna, Cepopeda, Caldocera and Rotifers [27,28,29].

### 2.3. Benthos

Benthic macro-invertebrate organisms are those that are found interred in the sand, attached to rocks or those that walk on the bottom, such as clams and cockles, mussels or crabs. Mussels have been recognised as ideal sentinels for the assessment of aquatic pollution because they have a wide geographical distribution, are easy to collect and are filter-feeders that accumulate pollutants in their bodies [32]. In addition, they have a long life-cycle, which allows the study of the effects of pollution over a long period of time [33]. However, although these organisms have often been used as bioindicators of marine pollution, pharmaceutical bioaccumulation is poorly developed, and the presence of these compounds in benthic species differs between sampling sites. Some authors have proposed the used of caged organisms rather than in wild ones, as it varies between species and the distribution and abundance of these specimens’ changes spatially and temporally [34]. In the literature, the most studied molluscs were bivalves, specifically mussels, but also oysters, clams, limpets and sea snails [35,36,37]. Other molluscs also studied have been gastropods (conch, snail) and cephalopods, such as octopus [38,39,40]. Oher benthos organisms such as crustaceans and echinoderms were studied for the determination on pharmaceuticals in aquatic environments, such as starfish as echinoderm [37] and barnacles, shrimp and crabs as crustaceans [27,38,41]. The most studied drugs include the antibiotic sulfamethoxazole, the analgesic naproxen, the antiepileptic carbamazepine and the antidepressant venlafaxine [37,42,43,44].

### 2.4. Fish

Fish are considered one of the most important bioindicators in both fresh and salt waters to estimate the level of pollution in the environment [3]. They have the ability to accumulate pollutants present in the surrounding environment in their fatty tissues [45]. Biomonitoring of these species is important due to human consumption, as they are a higher link in the food chain and, besides the inhalation exposure, the presence of contaminants in their bodies may be due to biomagnification (dietary exposure). Human exposure is the main reason to study the bioaccumulation of PhACs in different fish species as well as other biota across trophic levels [34]. Among the different fish species ussually used in bioaccumulation studies are carp [38], flatfish [43], salmon and rainbow trout [46] or mullet [47,48]. Regarding the PhACs studied, they belong to many families of drugs, including antibiotics such as quinolones, sulphonamides, and tetracyclines [39,40], analgesics such as naproxen, diclofenac, and acetaminophen [49,50,51] and other families such as antidepressants, β-blockers or antiepileptics [52,53].

## 3. Analytical Methodologies for the Determination of Pharmaceuticals in Biota Samples

The growing concern about the contamination of the environment has led to an increase in the number of publications focused on the detemination of PhACs in aquatic organisms in recent years. Table 1a–i summarizes the most relevant methods from the analytical point of view classified by taxonomic groups. Additionally, the graphs shown in Figure 2 represent the extraction techniques (a) and clean-up procedures (b) most commonly used for the sample treatment in the reviewed articles.

### 3.1. Sample Collection

In most of the literature consulted, specimens were captured by professional divers in different sampling areas, although in some cases, they were purchased in local supermarkets, either to be used as an analyte-free matrix [53], or as study sample [64,74,75]. Once captured, they were transported on ice, in order to avoid decomposition, at −10 °C and stored frozen at −20 °C [53,72,77], or deep-frozen (around −80 °C) until analysis [64,73,78].

### 3.2. Sample Pretreatment

Prior to storage, in order to guarantee the homogeneity of the sample as well as to reduce the particle size, and therefore, to achieve better extraction efficiency, most of the articles consulted pulverized the sample. The fish were cleaned before spraying the specimens and then, according to the literature, the vast majority of studies homogenised the sample by analysing a pool of all the body cavities of the different fish. However, in some cases, fish were deboned [80], only the muscle was analysed [39,42,46,69,78] or the different body cavities were analysed separately (fillet, gills, liver, intestine or brain) [49,53,56,82]. In addition, usually the samples were freeze-dried, so that spraying in the absence of humidity would be easier, although several studies worked with wet weight [50,79]. In case of molluscs, cephalopods or crustaceans were generally pooled without differentiating body cavities, removed from the shell if present, freeze-dried and ground into powder [34,60,68]. Ojemaye and Petrick [36], for the study of algae and echinoderms, rinsed, shelled and dissected by freeze-drying, and in the case of plankton, they were washed, homogenized and stored at −20 °C [37,62]. Storage consisted of frozen maintenance until analysis at −20 °C.

### 3.3. Sample Treatment (Extraction and/or Clean-Up)

#### 3.3.1. Ultrasound USE and FUSLE

An ultrasound consists of a mechanical wave propagation that is formed by cycles of compression and refraction, that is, waves of high and low pressures combined. The wave frequencies are above 20 kHz. Ultrasonic solvent extraction (USE) is able to induce these compressions and refractions of solvent molecules resulting in the formation of bubbles due to temperature and pressure variations. Collisions between particles as well as ultrasonic waves are able to induce fragmentation, which reduces the particle size, helping the mass transfer. The implosion of bubbles on the matrix surface results in erosion, which improves solvent accessibility [56]. Ultrasonic irradiation can be indirect or direct, both of which will be explained below. Argüello-Pérez et al. [77] determine four analgesics in fourteen different fish species using USE at 20 °C at 400 W power with a surface area of 3.8 cm^2^, achieving recoveries close to 100% in all cases. Focused ultrasound solid-liquid extraction (FUSLE) is a relatively new extraction technique, which started gaining popularity because the ultrasonic bath often provides low power. By introducing a probe directly into the extraction mixture, a sonication power up 100 times higher is achieved, as well as greater reproducibility and efficiency. The ultrasound energy is concentrated at the tip of the probe and is hence focused [83], and when ultrasound waves cross the liquid, many gaseous bubbles are formed which, when they implode, produce locally very high temperatures as well as high pressures and velocities of solvent micro-jets [84]. Mijangos et al. used FUSLE to extract antibiotics, analgesics and antiepileptics, among others from mussels and sea bream. For the extraction, authors used 30 s and 10% amplitude with 7 µL of MeOH/H_2_O (95:5, *v*/*v*) as solvent at 0 °C (extraction efficiencies from 71 to 126%) [85]. Some works apply ultrasound in a simpler way, by sonication in a common laboratory ultrasound machine. In this case, ultrasonic irradiation takes place indirectly, i.e., through the sample container. This equipment works at a single frequency, therefore the wave amplitude cannot be controlled. Danesaki et al. [74] used an ultrasonic bath at 60 °C (20 min) followed by a precipitation of lipids and proteins to recover 143 veterinary drugs from fish, while Ali et al. [51], analyzed different PhACs at room temperature (15 min) obtaining recoveries between 30% and 103% and limits of detection (LODs) from 0.1 to 13 ng mL^−1^.

#### 3.3.2. Pressurized Liquid Extraction

Pressurized liquid extraction (PLE), also called accelerated solvent extraction (ASE), is used for the extraction of analytes from solid or semi-solid matrices, by combining the use of different solvents with high temperatures and pressures. This allows higher recoveries and good extraction efficiencies while decreasing extraction time [65]. MeOH, acetonitrile (ACN) and water, or a mixture of them, have frequently used as extractant solvents. In addition, working temperatures are around 50 °C. Rojo et al. [52] studied different families of PhACs in fish muscle tissue, achieving recoveries between 26 and 115%. Other authors have proposed this technique to investigated different drugs in several types of fish as well as biofilm, plankton, bivalves, crustaceans and cephalopods obtaining LODs between 0.0004 and 6 ng g^−1^ and recoveries ranging from 20 to 151% [26,34,39,47,75,86].

#### 3.3.3. Microwave Assisted Extraction

Microwave-assisted extraction (MAE) was first used to replace Soxhlet extraction with the aim of reducing the amount of extraction solvent, achieving similar or better recoveries than Soxhlet extraction and reducing digestion time. It consists of heating the closed vessel to warm the solvent and decrease its viscosity, while increasing the solubility of the analytes in the extraction solvent and to facilitate the penetration into the matrix [54]. In the literature consulted, only the research by Argüello-Pérez et al. used this assisted extraction technique, for the analysis of several antimicrobials in fish as matrix [72]. ACN was used as solvent and it was carried out for 5 min at 40 °C with a power of 400 W. They obtained recoveries higher than 87% for all analytes and LODs between 4.54 and 101.3 pg kg^−1^.

#### 3.3.4. Solid-Phase Extraction

Solid phase extraction (SPE) allows the concentration of a target analyte by removing interferents present in the matrix via a solid stationary phase. This is an absorbent, which will be chosen according to the physicochemical properties of target compounds, in order to correctly separate the analytes from the rest of the interferents [76]. There are different types of sorbents; some of them retain the analytes and others the inteferents.

Boulard et al. [61] used silica gel for cleaning fish liver and fillet extracts in bream together with water and ACN to remove non-polar compounds from the extract. They achieved low LODs for the different PhACs, between 0.05 and 5.5 ng mL^−1^. Another sorbent used in SPE is the alumina column, which is capable of retaining compounds with an acidic character. It is used for the separation of compounds with medium polarity [87]. Huang et al. [72] used an alumina column in the clean-up phase for the determination of 6 antibiotics in fish muscle. This clean-up took place in two steps, after the alumina column in which ACN was used; a DLLME was carried out. They achieved recoveries higher than 87%.

According to the scientific literature consulted, SPE with cartridges is the most commonly cleun-up technique. Among all the sorbents, the most widely used cartridge is the HLB, as it is a universal for acidic, neutral or alkaline compounds. Other sorbents packed in the cartridges are SAX and PSA, which are multilayer cartridges suitable for polar interactions. Chen et al. used this type combined with the HLB cartridge, facilitating the separation of polar and non-polar compounds for sulfonamides and tetracyclines in crabs, shrimps and different types of fish, reaching recoveries between 50 and 150% [41]. McEneff et al. used a cartridge with Strata-X, which is a reversed-phase polymeric cartridge, at SPE for the determination of different analgesics and antiepileptic drugs in mussels, achieving yields between 83 and 94% [60]. Tanoue et al. used a Hybrid SPE-Phospholipid cartridge, which removed exogenous proteins as well as phospholipid interferences for different drugs and some of their metabolites in fish analysis, with recoveries between 70 and 120% [49].

Gao et al. developed a different type of clean-up based on SPE [79]. These authors used a metal organic framework (MOF) as adsorbent. SPE (CF@UiO-66-NH_2_) is a MOF based on Zr and modified with cotton fiber, resulting in CF@UiO-66-NH_2_, which has a high adsorption capacity because it has many active sites. After adsorption, desorption of the analytes takes place by using desorption solvents. Gao et al. [79] used this adsorbent for the extraction of some analgesics such as ketoprofen, naproxen, flurbiprofen, diclofenac sodium and ibuprofen in fish and crustaceans’ tissue, achieving recoveries between 95 and 116.99% and LODs between 0.12 and 3.50 ng mL^−1^.

#### 3.3.5. Dispersive Solid Phase Extraction (dSPE)

This technique consists of the dispersion of a solid sorbent in a liquid or dissolved sample so that impurities or interferents are retained, resulting in a clean extract. After separation, the sorbent is removed, usually by centrifugation [88]. There are different types of sorbents; those used in the consulted literature will be explained below.

C18 sorbent is used for the extraction of non-polar or relatively polar compounds, being able to retain most of the organic compounds present in an aqueous phase.

QuEChERs (quick, easy, cheap, effective, rugged and safe) is one of the most user-friendly techniques. High extraction efficiencies can be achieved and it is also in agreement with green chemistry as it uses a small amount of sample as well as solvent. This makes it one of the most widely used extraction methods nowadays [89]. This technique is applied in two sequential stages. The first one is the extraction phase, which is performed using an organic solvent, normally ACN in the presence of different salts, such as MgSO_4_ or NaCl, whose function is to regulate pH, control polarity to favor the phase separation and contribute to the recovery of the analyte. Then, a second stage of cleaning is carried out, which consists of purification dSPE. With this step, the residual water and other interfering compounds present in the matrix are removed. For this purpose, some salts are used, such as MgSO_4_, which removes excess water; PSA (primary/secondary amine), which removes organic acids, fatty acids and sugars from the matrix; C18 (sorbent), which eliminates fats and other non-polar interferences; and graphitized black carbon (GCB), which removes pigments from the sample [89].

#### 3.3.6. Others

Soxhlet. Since it involves much larger quantities of solvent and much longer times than other extraction techniques, and the yields of extraction obtained are not much better, it is a technique rarely used today. It consists of the continuous flow of solvent through the sample, using a distillation flask. When the solvent condenses, it does so with the dissolved analytes. This operation is repeated until extraction process is completed, achieving good extraction efficiencies [90]. Ojemaye and Petrick [36,48] used this technique for the extraction of a group of drugs, such an antiepileptic, antibiotics and an analgesic, in fish, bivalves, algae and echinoderms. They used MeOH and ACN (3:1, *v*/*v*) as extractant solvents and they achieved recoveries between 69.2 and 107.5% for fish and 96.1 and 100.5% for the rest of the species as well as LODs of 0.01 and 0.036 ng g^−1^ for fish and between 0.62 and 1.05 ng L^−1^ for the other species under study.

TissueLyser II. TissueLyser consists of bead mill equipment which, with adapters, is capable of lysing biological samples by agitation at high speeds. It has many applications, such as the disruption of human, animal, plant and even bacterial tissues. It is a very efficient extraction [91]. Borik et al. [53] used this type of lysis for the extraction of citalopram from rainbow trout fish brain tissue, achieving close to 100% recovery with a LOD of 0.39 ng g^−1^.

Mechanical shaking. This is one of the simplest extraction techniques as it consists of stirring the sample with the extraction solvent for a certain time to ensure the migration of the analytes from the solid phase to the liquid one. Generally, this agitation is followed by centrifugation so that the decantation can take place and the phases can be separated correctly, leaving the target analytes dissolved in the liquid [92,93]. Not many studies based on the use of this technique have been found, as the time required is usually longer. The most commonly used solvents are can and MeOH, sometimes acidified with formic or acetic acid. López-García et al. [61] used ACN with salts (MgSO_4_, NaCl, sodium citrate and DCS (sodium citrate sesquihydrate)) for the study of mussel’s tissue, with recoveries between 77% and 118% and and low LODs (<2 ng g^−1^). Bobrowska-Korczak et al. [64] and Miossec et al. [73], studied the presence of 98 and 41 PhACs, respectively, in fish and shrimps, with LODs between 0.1 and 40.2 ng g^−1^, reaching recoveries in the range of 28 to 188%.

Cell disruption. This technique is carried out in a high-speed shaking equipment that, in a very short time, is able to extract the maximum amount of DNA, RNA, proteins and other compounds with very good efficiency. This is why after this type of extraction the cleaning and purification protocol plays an essential role in the removal of interferents. Boulard et al. [61] used this extraction technique for the analysis of 26 PhACs in bream and the time required for extraction was 40 s, achieving recoveries from 70% to 130% and LODs from 0.05 to 5.5 ng mL^−1^.

Pulverised liquid extraction (PuLE). In this extraction technique, the sample is homogenized and the analytes are extracted simoultaneously by shaking. The solid sample is placed in a vessel together with two glass beads and then it is agitated in a homogeniser at a known speed and time. Only one study found in the scientific literature have used this extraction modality. This technique was used to extraxt 29 PhACs in the amphipod *Gammarus pulex*. The recoveries were between 41 and 89% [70].

Gel permeation Chromatography (GPC) is a technique traditionally used for the clean-up of the extracts because it removes biological macromolecules such as fats or proteins, separating them according to size. The column packing is a porous gel, and the beads packaged in it interact with the compounds, so it differs from other separation techniques in that it does not rely on physical or chemical interactions [94]. Rojo et al. used GPC for clean-up of the extracts of fish species when they had determined 15 PhACs and two of their metabolites, achieving recoveries between 26 and 115% [52]. Álvarez-Muñoz et al. studied 8 PhACs from different families in 9 different fish species using GPC as a clean-up technique [75].

Of all the extraction techniques described in this section, those based on the use of ultrasound (USE and FUSLE) have been the most attractive alternatives for the analysis of PhACs in biota (36% of the studies), followed by PLE (30% of the consulted studies). Both techniques are simple, provide automatization, short extraction times and low solvent consumption. For clean-up, SPE using Oasis HLB cartridges has been shown to be an efficient method and the most popular used as a clean-up procedure (71% of the studies), regardless of the aquatic organism under study.

## 4. Instrumental Analysis

### 4.1. Liquid Chromatography

LC separation technique coupled with an adequate detector allows quantitative determinations of the compounds with high selectivity, sensibility and accuracy. LC is a very suitable technique for the multiresidue PhACs separation. Furthermore, it does not require the previous derivatization step.

Regarding the retention mechanisms, a broad variety may be applicable in LC. Some examples are reverse phase chromatography (RP-LC), normal phase liquid chromatography (NP-LC), hydrophilic interaction liquid chromatography (HILIC), ion-pairing chromatography (IPC), ion exchange chromatography (IEC), or hydrophobic interaction chromatography (HIC), among others. As far as the determination of PhACs in aquatic organisms is concerned, and considering the physicochemical properties of the target compounds (polar compounds), the RP-LC modality has been the best choice for all the authors. This retention mechanism is related to non-polar selectivity consisting of a non-polar stationary phase and, as mobile phases, a solvent mixture of high polarity solvents. Consecuently, the least polar compounds of the mixture appear first in the chromatogram. RP-LC using C18 silica columns is mainly used for separation, although chiral columns based on α1-glycoprotein (AGP) and phenyl or phenyl-hexyl columns have been also used as stationary phases [44,73,78]. Generally, the most commonly used solvents in the mobile phase are water as the aqueous component (phase A), and in the organic phase, ACN or MeOH (phase B) [68,78,79]. Some authors such as Moreno-González et al. used dichloromethane and methanol (90:20, *v*/*v*) in isocratic mode as mobile phases for the analysis of 20 PhACs in fishes and molluscs, prior to a study of bioaccumulation [47]. Sometimes, the use of additives in the aqueous phase, or occasionally in both, such as formic acid, ammonium formate, ammonium acetate or acetic acid at low concentrations, assists ionization when mass spectrometry is selected as detection technique. The use of additives provides better analytical signals and thus, make it easier to determine the target analytes [52,70,73,95].

On the other hand, HILIC is considered by far an attractive alternative for the separation of polar compounds, such as pharmaceuticals. This one is associated to polar selectivity, but also using polar mobile phases. Although the reported articles were based on RP-LC, the use of diol and amine columns may be also considered, as they could provide promising results in the separation of PhACs.

In recent years, the HPLC technique has been largely replaced by UHPLC as it has many advantages over the former. The analyses are faster and more sensitive. This is due to the fact that the column packing consists of smaller and more porous particles (sub-2-micron particles) that achieve better chromatographic peaks, and therefore greater sensitivity, although the collateral effect is that the work is carried out at higher pressures. As this review work has focused on the last 10 years of research, most of the studies included the use of UHPLC technique [34,46,72] (56%) while the remaining 44% used classical HPLC (Table 1a–i). The chromatographic columns used in the first case are usually 10 cm long [13,27], although some studies achieve separation even with 5 cm columns [35,39,60]. In the case of HPLC, longer chromatographic columns are used, usually 15 cm [28,55,56,58,62], with the exception of some studies using shorter columns of 10 cm [42,52] or 12.5 cm in length [61].

### 4.2. Detection Systems

After chromatographic separation, spectrophotometric detection has been used on a limited, but interesting, number of cases, depending on the properties of the compounds under study [96]. For example, Gao et al. coupled an ultraviolet detection system for the determination of 5 NSAIDs in fish and shrimp muscle tissues using a new synthetic MOF in the extraction of the compounds, achieving LODs between 0.12 and 3.50 ng mL^−1^ [79]. It is a universal and inexpensive detector that is very useful for routine analysis.

However, MS was the most common detection system used in the literature consulted. For the ionization of the sample, the main interface used is electrospray ionization (ESI). In the literature consulted, 80 studies indicate the use of this interface. ESI involves generating ions by applying a high voltage to a liquid, generating an aerosol. It is often used in the case of macromolecules, as they tend to fragment after ionization. Other interfaces used are atmospheric pressure chemical ionization (APCI) [75] and heated electrospray ionization [63]. In both cases, they use heat and a nebulization gas to form an aerosol and ionize the molecules in the gas phase. In some cases, thermal degradation may occur due to the use of heat, so this interface is often used when the analytes are heat stable and volatile. For that reason, articles consulted in the literature mainly used ESI as an interface, as the PhACs are generally high molecular weight compounds [71].

Based on MS resolution, two main categories are typically distinguished: low resolution (LRMS) and high resolution (HRMS) mass spectrometry. The former gives two decimal m/z digits and is commonly used in targeted analysis, while the latter offers higher resolving power which is advantageous in non-targeted analysis. In the reviewed works, LRMS, in particular, tandem mass spectrometry (MS/MS) using a triple-quadrupole mass analyzer (QqQ), is the most frequently used because of its increased selectivity, low LODs and improved S/N ratio. Multiple reaction monitoring mode (MRM) is particularly useful for the simultaneous determination of different classes of PhACs in one single run and has been able to detect large amounts of analytes in complex matrices even in trace quantities [46,51,68,85]. López-García et al. [61] used a QqLit analyzer (quadrupole ion trap), consisting of three quadrupoles analyzers in which the last one acts as a linear ion trap, offering better sensitivity. In the determination of psychoactive substances in mussels, they achieved LODs below 2 ng g^−1^, with high recoveries. Similarly, the use of other systems based on MS/MS, as the HCT (ultra ion trap) [80] and the QTRAP mass spectrometer [48,50,74], have been proposed. In contrast, it should be noted that only one study used a simple quadrupole analyzer. They determined three drugs in sea sponge, achieving detection limits between 0.01 and 10 ng g^−1^ with a recovery of 80% [70].

Likewise, the HRMS counterpart has undergone a noteworthy evolution in the last years. Although it is typically used in non-targeted analyzes when the compounds are unknown a priori, it has been shown to possess sufficient resolving power for quantitative purposes as well. It is especially useful for example to know the transformation products or identify compounds with the same molecular mass, thanks to the structural fragmentation patterns, the accurate mass, and the isotopic distribution. In light of this, analyzers such as Orbitrap or TOF, which also offer very good characteristics, have been employed in some of the revised works [41,48,59,63,73,77,79]. For example, Baesu et al. and Danesaki et al. used the Q-TOF for the determination of drugs from different families in fillet of fish, reaching LODs of 0.2–2.6 ng g^−1^ and 20–200 ng g^−1^, respectively [73,77]. Kalogeropoulou et al. used a Q-Orbitrap MS achieving limits of quantification (LOQs) between 0.5–19 ng g^−1^ for the analysis of several antibiotics, antiepileptics and antidepressants in fish muscle [79].

## 5. Conclusions and Future Perspectives

Advances in analytical tools and instrumentation have allowed the development of a high number of sensitive and selective methods to determinate a broad range of PhACs in complex matrices, such as aquatic organisms. The present work provides an overview of the recent available methodologies for the analysis of PhACs in aquatic biota from different levels of the food chain. Among the PhACs, most investigated were antibiotics (ciprofloxacin, trimethoprim, and sulfamethoxazole), non-steroidal anti-inflammatory drugs (NSAIDs), analgesics (diclofenac, ibuprofen, naproxen and acetaminophen), antidepressants (venlafaxine) and antihypertensive drugs (propranolol and metoprolol), in this order, which also corresponds to those most accepted and consumed by the human population. Other groups, such as the cholesterol-lowering, antidiabetic and anticancer drugs, which have greatly increased in the last decade, have occasionally been considered in the studies consulted [97]. In addition, it should be noted that limited research has been conducted to analyze their transformation products (metabolites and degradation products) which emphasizes the need to develop analytical methods to cover this gap.

In relation to the studied taxonomic groups in the determination of PhACs, fish has been the most extensively organism investigated (33%), followed by molluscs (29%) and crustaceans (17%). In contrast, there are few proposed methods to assess the presence of these compounds in echinoderms (1%), and in biota of the first level of the food chain such as algae (2%), phytoplankton (5%), or zooplankton (8%). Therefore, more studies are needed to analyze PhACs at the lowest levels of the food chain, such as producers and benthic primary consumers, since the latter seem to be the main bioaccumulators for filter-feeding [98]. This would help to broaden the knowledge about the trophic transfer of PhACs, a barely explored field.

Given the complexity of biota matrices, special attention has been played to the sample preparation step, both extraction and purification, to obtain clean extracts and not compromise instrument sensitivity due to matrix effects. The extraction step is key in determining the analytical parameters of the method. As far as extraction techniques, extraction using ultrasound (USE, FUSLE) has been the most attractive alternative, used in 36% of the studies consulted, followed by PLE, used in 30% of the studies. Both techniques provide automatization, short extraction times and low solvent consumption, compared to other techniques, such as traditional Soxhlet extraction. ACN, MeOH and water have been the solvents of choice for UAE while for PLE, in addition to these, the combination of acetone and MeOH has been extensively used. However, other green techniques should be explored for the extraction of these compounds to further reduce solvent and extraction time, such as aqueous two-phase systems (ABS), which remove volatile organic compounds and have very promising prospects. For clean-up, SPE using cartridges has shown to be an efficient method and the most popular used as a clean-up procedure (71% of the studies), regardless of the aquatic organism under study. Polymeric reversed-phase sorbents, and in particular Oasis HLB cartridges, have been the most suitable par excellence. Future trends in PhACs analysis in biota may include the design of on-line extraction techniques to reduce sample handling and avoid tedious sample treatments.

Finally, UHPLC-MS/MS has shown to be the most widely used technology for the analysis of PhACs due to the benefits it can offer. On the one hand, there has been a trend towards the use of UHPLC since, unlike HPLC, it operates at higher pressures and provides better resolution due to shorter column lengths and smaller particle sizes. On the other hand, its coupling with MS/MS detection is advantageous as it provides high sensitivity and selectivity, allowing quantification in the low ng L^−1^ or ng g^−1^. It should also be noted that some recent works, instead, have used HRMS (Orbitrap or QTOF analyzers) for determining PhACs in the organisms under study being able to distinguish between compounds with comparable masses.

## Figures and Tables

**Figure 1 molecules-27-07569-f001:**
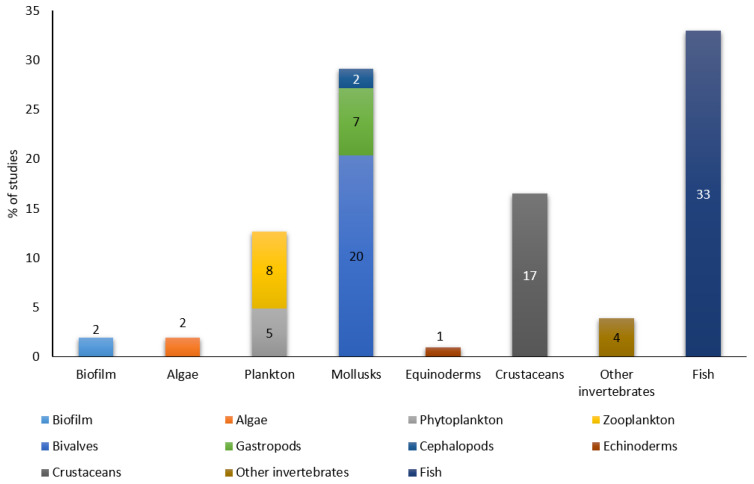
Number of consulted studies according to the different taxonomic groups in aquatic environment.

**Figure 2 molecules-27-07569-f002:**
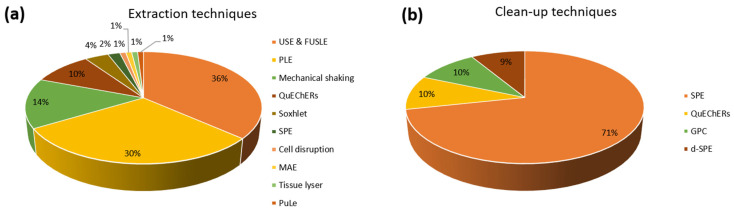
Extraction techniques (**a**) and clean-up treatments (**b**) used in the reviewed publications.

**Table 1 molecules-27-07569-t001:** (**a**) Analytical methods performance for PhACs concentration determination in biofilm. (**b**) Analytical methods performance for PhACs concentration determination in algae. (**c**) Analytical methods performance for PhACs concentration determination in plankton. (**d**) Analytical methods performance for PhACs concentration determination in molluscs. (**e**) Analytical methods performance for PhACs concentration determination in cephalopods. (**f**) Analytical methods performance for PhACs concentration determination in echinoderms. (**g**) Analytical methods performance for PhACs concentration determination in crustaceans. (**h**) Analytical methods performance for PhACs concentration determination in other invertebrates. (**i**) Analytical methods performance for PhACs concentration determination in fish.

(a)									
Specie	Pharmaceuticals	Type and Amount of Sample (g)	Pre-Treatment	Treatment		Analysis	Analytical Features		Ref.
				Extraction Technique	Clean-Up		Recovery (%)	LOD (ng g^−1^)	
No data	Diclofenac, ibuprofen, 1-OH-ibuprofen, piroxicam, diltiazem, propyphenazone, sulfamethoxazole, verapamil, norverapamil, hydrochlorothiazide, bezafibrate, gemfibrozil, pravastatin, carbamazepine, acridone, 10,11-epoxy-CBZ, 2-OH-CBZ, citalopram, fluoxetine, paroxetine, venlafaxine, azaperone, dexamethasone, metoprolol, propanolol	0.2 (d.w)	Freeze-dried, stored at −20 °C	PLE (citric buffer (pH 4)/ACN)	No data	UHPLC-MS/MS	No data	No data	[54]
Periphyton (No data)	Ethinylestradiol, acetaminophen, diclofenac	0.67 (d.w)	Air dry, powdered	USE (ACN/ MeOH 1% acetic acid)	No data	HPLC-MS/MS	62	No data	[55]
**(b)**									
**Specie**	**Pharmaceuticals**	**Type and Amount of Sample (g)**	**Pre-Treatment**	**Treatment**		**Analysis**	**Analytical Features**		**Ref.**
				**Extraction** **Technique**	**Clean-Up**		**Recovery (%)**	**LOD** **(ng g^−1^)**	
Sea lettuce (*Ulva* sp.), Red algae (*Gelidium pristoides*), Hanginng wrack (*Bifurcaria brassicaeformis*), Strap caulerpa (*Caulerpa filiformis*), Slippery orbits (*Aeodes orbotisa*)	Phenytoin, lamivudine, acetaminophen, caffeine, sulfamethoxazole, diclofenac, carbamazepine	10 (d.w)	Rinsed, deshelled and dissected. Freeze-dried	Soxhlet (MeOH/ Acetone)	SPE (HLB cartridges)	UHPLC-MS/MS	96.1–100.5	0.62–1.05 ng L^−1^	[37]
Water starwort (*Callitriche* sp.), Pondweed (*Potamogeton* sp.)	Ethinylestradiol, acetaminophen, diclofenac	0.5 (d.w)	Air dry, powdered	USE (ACN/ MeOH 1% acetic acid)	No data	HPLC-MS/MS	81	No data	[55]
**(c)**									
**Phytoplankton**									
**Specie**	**Pharmaceuticals**	**Type and Amount of Sample (g)**	**Pre-Treatment**	**Treatment**		**Analysis**	**Analytical Features**		**Ref.**
				**Extraction** **Technique**	**Clean-Up**		**Recovery (%)**	**LOD** **(ng g^−1^)**	
No data	Roxithromycin, erythromycin, ofloxacin, norfloxaxin, ciprofloxacin, tetracycline, sulfamethoxazole, sulfadiazine, sulfaquinoxaline, ibuprofen, diclofenac, naproxen, bezafibrate, propranolol, ketoconazole, carbamazepine, caffeine, sertraline, fluoxetine, norfluoxetine, citalopram, paroxetine, venlafaxine, duloxetine, bupropion, amitriptyline, clozapine, fluvoxamine, quetiapine, aripiprazole, chlorpromazine	0.5 (d.w)	Freeze-dried, homogenized, stored at −80 °C	PLE (MeOH/ acetone)	SPE (HLB cartridges)	UHPLC-MS/MS	66–128	0.07–1.67	[27]
No data	Sulfadiazine, suldapyridine, sulfacetamide, sulfamethazine, sulfamethoxazole, trimethoprim, norfloxacin, ciprofloxacin, ofloxacin, lomefloxacin, oxytetracycline, tetracycline, erythromycin, roxithromycin	0.1–0.5 (d.w)	Freeze-dried, stored at −18 °C	QuEChERs (ACN, acetic acid, 0.1 M EDTA, NaCl, Na_2_SO_4_)	d-SPE: QuEChERs (ACN, PSA, C18, Na_2_SO_4_)	LC-MS/MS	80.3–104.9	0.04–0.1	[28]
Cyanobacteria (*Microcystis aeruginosa*), Chlorophyceae (*Pediastrum* spp. *Crucigenia* spp. *Scenedesmus* spp.), Sensu lato (*Coscinodiscus* spp., *Cyclotella* spp.), Diatoms (*Melosira* spp., *Aulacoseira* spp.), Dinophycaeae (*Peridiniopsis* spp.), Cryptophyceae (*Cryptomonas*), Chrysophyceae (*Dinobryon* spp.), Euglenoidea (*Euglena* spp.)	Sulfadiazine, sulfapyridine, sulfacetamide, sulfamethoxazole, sulfamethazine, trimethoprim, lomefloxacin, ciprofloxacin, norfloxacin, oxytetracycline, tetracycline, roxithromycin, dehydroerythromycin	0.5–1.0 (d.w)	Frozen at −80 °C, stored in a vacuum desiccator	PLE (MeOH/ acetone)	SPE (HLB cartridges)	UHPLC-MS/MS	No data	No data	[29]
Cyanobacteria (No data)	Sulfachlorpyridazine, sulfadiazine, sulfadoxine, sulfamerazine, sulfadimethoxine, sulfamethazine, sulfamethoxazole, sulfamonomethoxine, sulfapyridine, sulfaquinoxaline, sulfisoxazole, sulfathiazole, trimethoprim, chlortetracycline, doxycycline, oxytetracycline, tetracycline, ciprofloxacin, difloxacin, danofloxacin, enrofloxacin, fleroxacin, lomefloxacin, marbofloxacin, norfloxacin, ofloxacin, pefloxacin, sarafloxacin, azithromycin, clarithromycin, leucomycin, oleandomycin, roxithromycin, tylosin, salinomycin, monensin, florfenicol, chloramphenicol	1 (d.w)	Washed (water), freeze-dried, stored at −20 °C	USE (MeOH, sodium acetate buffer pH 4)	SPE (SAX, HLB cartridges)	RRLC-MS/MS	54.2–117	0.02–9.38	[38]
Green algae (*Chlorophyta*), Diatoms (*Bacillariophyta*), Blue green algae (*Cyanophyta*)	Roxithromycin, erythromycin, ofloxacin, norfloxacin, ciprofloxacin, tetracycline, chloramphenicol, sulfamerazine and sulfadiazine, sulfamethoxazole, ibuprofen, diclofenac, naproxen and indomethacin, clofibric acid, gemfibrozil and bezafibrate, 17β-estradiol, 17α-ethynylestradiol, propranolol, carbamazepine, ketoconazole, sertraline	0.25 (d.w)	Freeze-dried, ground, stored at −20 °C	PLE (MeOH/ acetone)	SPE (HLB cartridges)	LC-MS/MS	68–116	0.01–1.12	[56]
**Zooplankton**									
**Specie**	**Pharmaceuticals**	**Type and Amount of Sample (g)**	**Pre-Treatment**	**Treatment**		**Analysis**	**Analytical Features**		**Ref.**
				**Extraction** **Technique**	**Clean-Up**		**Recovery (%)**	**LOD** **(ng g^−1^)**	
Water flea (*Daphnia magna*)	Roxithromycin, propanolol	Each sample point consisted by 10 daphnia individuals	Homogenized	Sonication (ACN)	SPE (HLB cartridges)	UHPLC-MS/MS	83–106	0.2	[57]
Water flea (*Daphnia magna*)	Tetracycline	30 organisms	Homogenized	MeOH, formic acid, EDTA	No data	LC-MS/MS	84.23	0.31 µg L^−1^	[58]
No data	Sulfachlorpyridazine, sulfadiazine, sulfadoxine, sulfamerazine, sulfadimethoxine, sulfamethazine, sulfamethoxazole, sulfamonomethoxine, sulfapyridine, sulfaquinoxaline, sulfisoxazole, sulfathiazole, trimethoprim, chlortetracycline, doxycycline, oxytetracycline, tetracycline, ciprofloxacin, danofloxacin, difloxacin, enrofloxacin, fleroxacin, lomefloxacin, marbofloxacin, norfloxacin, ofloxacin, pefloxacin, sarafloxacin, azithromycin, clarithromycin, leucomycin, oleandomycin, roxithromycin, tylosin, salinomycin, monensin, florfenicol, chloramphenicol	1 (d.w)	Washed (water), freeze-dried, stored at −20 °C	USE (MeOH, sodium acetate buffer pH 4)	SPE (SAX, HLB cartridges)	RRLC-MS/MS	54.2–117	0.02–9.38	[38]
No data	Roxithromycin, erythromycin, ofloxacin, norfloxaxin, ciprofloxacin, tetracycline, sulfamethoxazole, sulfadiazine, sulfaquinoxaline, ibuprofen, diclofenac, naproxen, bezafibrate, propranolol, ketoconazole, carbamazepine, caffeine, fluoxetine, norfluoxetine, citalopram, paroxetine, sertraline, venlafaxine, duloxetine, bupropion, amitriptyline, fluvoxamine, clozapine, quetiapine, aripiprazole, chlorpromazine	0.5 (d.w)	Freeze-dried, homogenized, stored at −80 °C	PLE (MeOH/ acetone)	SPE (HLB cartridges)	UHPLC-MS/MS	66–128	0.07–1.67	[27]
Copepoda, Cladocera, Rotifera (No data)	Sulfadiazine, suldapyridine, sulfacetamide, sulfamethazine, sulfamethoxazole, trimethoprim, norfloxacin, ciprofloxacin, ofloxacin, lomefloxacin, oxytetracycline, tetracycline, erythromycin, roxithromycin	0.1–0.5 (d.w)	Freeze-dried, stored at −18 °C	QuEChERs (ACN, acetic acid, 0.1 M EDTA, NaCl, Na_2_SO_4_)	d-SPE: QuEChERs (ACN, PSA, C18, Na_2_SO_4_)	LC-MS/MS	81.1–100.7	0.01–0.12	[28]
Copepoda, Cladocera, Rotifera (No data)	Roxithromycin, erythromycin, ofloxacin, norfloxacin, ciprofloxacin, tetracycline, chloramphenicol, sulfamerazine, sulfadiazine, sulfamethoxazole, ibuprofen, ketoconazole, diclofenac, naproxen, indomethacin, clofibric acid, gemfibrozil, bezafibrate, 17β-estradiol, sertraline, propranolol, 17α-ethynylestradiol, carbamazepine	0.25 g (d.w)	Freeze-dried, ground, stored at −20 °C	PLE (MeOH/ acetone)	SPE (HLB cartridges)	LC-MS/MS	68–116	0.01–1.12	[39]
Copepoda, Cladocera, Rotifera (No data)	Sulfadiazine, sulfapyridine, sulfacetamide, sulfamethoxazole, sulfamethazine, trimethoprim, lomefloxacin, ciprofloxacin, norfloxacin, oxytetracycline, tetracycline, dehydroerythromycin, roxithromycin	0.5–1.0 (d.w)	Frozen at −80 °C, stored in a vacuum desiccator	PLE (MeOH/ acetone)	SPE (HLB cartridges)	UHPLC-MS/MS	No data	No data	[29]
No data	Nicotine, haloperidol, pyremethamine	0.14–0.2 (d.w)	Freeze-dried	USE (ACN, MeOH, H_2_O), vortex, USE	SPE (No data)	LC-HRMS/MS	70–130	0.05–5.70 *	[59]
Green algae (*Chlorophyta*), Diatoms (*Bacillariophyta*), Blue green algae (*Cyanophyta*)	Roxithromycin, erythromycin, ofloxacin, norfloxacin, ciprofloxacin, tetracycline, chloramphenicol, ibuprofen, diclofenac, naproxen, indomethacin, clofibric acid, sulfamerazine, sulfadiazine, sulfamethoxazole, gemfibrozil, bezafibrate, propranolol, carbamazepine, sertraline, ketoconazole, 17β-estradiol, 17α-ethynylestradiol	0.25 (d.w)	Freeze-dried, ground, stored at −20 °C	PLE (MeOH/ acetone)	SPE (HLB cartridges)	LC-MS/MS	68–116	0.01–1.12	[56]
**(d)**									
**Bivalves**									
**Specie**	**Pharmaceuticals**	**Type and Amount of Sample (g)**	**Pre-Treatment**	**Treatment**		**Analysis**	**Analytical Features**		**Ref.**
				**Extraction** **Technique**	**Clean-Up**		**Recovery (%)**	**LOD** **(ng g^−1^)**	
Oysters (*C. Gigas*), Clams (*C. gallina*), Mussels (*M. galloprovincialis*)	Ronidazole, metronidazole, dimetridazole, sulfamethoxazole, N-acetyl-sulfamethoxazole, azithromycin, erythromycin, venlafaxine, O-desmethylvenlafaxine, carbamazepine, 10,11-Epoxycarbamazepine, citalopram,2-Hydroxycarbamazepine, alprazolam, codeine, phenazone, propyphenazone, piroxicam, azaperone, azaperol, diltiazem, hydrochlorthiazide, tamsulosin	0.5 (d.w)	Shells removed, pooled for homogenizing, freeze-dried, ground and kept at −20 °C	PLE (MeOH/ H_2_O)	SPE (HLB cartridges)	UHPLC-MS/MS	40–115	0.01–0.80	[35]
Zebra mussels (*Dreissena polymorpha*)	Diclofenac	0.1 (d.w)	Freeze-dried and grinded	QuEChERs (H_2_O, ACN, heptane, acetate salt, DMSO)	d-SPE: QuEChERs (acetate salt)	UHPLC-MS/MS	73–117	0.02–1	[60]
Mussels (*Perna viridis*), Oysters (*Cassostrea hongkongensis*)	Acetaminophen, amitrimtyline, aripiprazole, benzoylecgonine, buprenorphine, caffeine, carbamazepine, diclofenac, diltiazem, diphenydramine, fluoxetine, methylphenidate, norfluoxetine, promethazine, sertraline, amlodipine, desmethylsertraline, trimethoprim, erythromycin, sucralose, sulfamethoxazole	1 (w.w)	Separated from their shells, homogenized and frozen at −20 °C	Mechanical shaking (0.1 M acetic acid, MeOH)	No data	LC-MS/MS	80–120	0.01–0.75	[36]
Mussel (*Mytilus galloprovincialis*)	Cocaine, benzoylecgonine, cocathylene, amphetamine, metamphetamine, MDMA, morphine, methadone, 6-monoacetylmorphine, EDDP, ketamine, lysergic acid diethylamide, A tetrahydrocannabinol, 11-hydroxy-A THC, 11-nor-9-carboxy-A THC, AH-7921, mephedrone, MDPV, caffeine, ephedrine, alprazolam, a-hydroxyalprazolam, midazolam, lormetazepam, a-hydroxymidazolam, diazepam, oxazepam, temazepam, citalopram, fluoxetine, sertraline, venlafaxine, zolpidem, chlorpromazine, hydroxyzine	10 (w.w)	Homogenized	Manual shaking (ACN, MgSO_4_, NaCl, NaCitrate, DCS)	d-SPE: QuEChERs (PSA, C18, MgSO4)	LC-MS/MS	77–118	<2	[61]
Mussel (*Mytilus* spp.)	Diclofenac, mefenamic acid, trimethoprim, carbamazepine, gemfibrozil	1 (d.w)	Freeze-dried, ground	PLE (ACN/H_2_O), Al_2_O_3_	SPE (Strata-X SPE cartridges)	LC-MS/MS	83–94	4–29 *	[62]
Mussel (*Mytilus galloprovincialis*)	Carbamazepine, oxcarbazepine + non target compounds (caffeine, metoprolol, cotinine, ketoprofen)	2 (d.w)	Freeze-dried	QuEChERs (ACN, Na_2_SO_4_, NaCl, Na_3_Cit: 2H_2_O, Na_2_HCit: 3H_2_O)	d-SPE: QuEChERs (Na_2_SO_4_, PSA, C18, formic acid)	LC-HRMS	67–110	0.1–0.3	[63]
Mussel (*Mytilus galloprovincialis*)	Diclofenac, diazepam, sotalol, carbamazepine, citalopram, venlafaxine, azithromycin, sulfamethoxazole	All edible meat (no data)	Pooled, homogenized, freeze-dried, kept at −20 °C	PLE (MeOH/ H_2_O)	SPE (HLB cartridges)	UHPLC-MS/MS	No data	0.01–0.65	[64]
Mussel (*Mytilus galloprovincialis*), Razor shell (*Ensis siliqua*), Cockle (*cerastoderma edule*)	Atenolol, metoprolol, nadolol, propanolol, sotalol, salbutamol, diazepam, carbamazepine, azaperol, azaperone, 10,11-epoxycarbamazepine, 2-OH-carbamazepine, citalopram, venlafaxine, alprazolam, chlorothiazide, codeine, phenazone, piroxicam, propyphenazone, ronidazole, dimetridazole, metronidazole, azithromycin, erythromycin	0.5 (d.w)	Freeze-dried	PLE (MeOH/ H_2_O)	SPE (HLB cartridges)	UHPLC-MS/MS	No data	0.01–2	[40]
Mussel (*Mytilus galloprovincialis*)	Trimethoprim, ciprofloxacin, norfloxacin, sulfadiazine, sulfamethoxazole, amitriptyline, clomipramine, imipramine, nortriptyline, eprosartan, irbesartan, losartan, diclofenac, telmisartan, valsartan, propanolol, acetaminophen, ketoprofen, bezafibrate, clofibric acid, carbamazepine, phenytoin	0.5 (d.w)	Freeze-dried, ground, homogenized	FUSLE (MeOH/ H_2_O)	SPE (HLB cartridges)	LC-MS/MS	71–126	4–48	[65]
Carib pointed-venus (*Anomalocardia brasiliana*), Blue Mussel (*Mytilus edulis*)	Bezafibrate, carbamazepine, chloramphenicol, diclofenac, 4′-Hydroxydiclofenac, furosemide, gemfibrozil, ibuprofen, indapamide, ketoprofen, naproxen, simvastatin	0.5 (d.w)	Dissection to obtain the morphometric measures, freeze-dried	QuEChERs (ACN, formic acid, NH_4_Cl)	QuEChERs (MgSO_4_, Z-Sep)	HPLC-MS/MS	77–126	0.002–1.09	[47]
Limpets (*Cymbula granatina and cymbula oculis*), Sea snail (*Oxystele sinensis and oxytele tigrina*), Mussel (*mytilus galloprovincialis*)	Phenytoin, lamivudine, acetaminophen, caffeine, sulfamethoxazole, diclofenac, carbamazepine	10 (d.w)	Rinsed, deshelled and dissected, freeze-dried	Soxhlet (MeOH/ Acetone)	SPE (HLB cartridges)	UHPLC-MS/MS	96.1–100.5	0.62–1.05 ng L^−1^	[37]
Oyster (*Ostrea gigas*), Scallop (*Mimachlamys nobilis*), Mussel (*Mytilus edulis*)	Sulfadiazine, sulfamerazine, sulfamethazine, trimethoprim, sulfamethoxazole, sulfathiazole, sulfapyridine, ciprofloxacin, norfloxacin, ofloxacin, tetracycline, flumequine, oxytetracycline, gemfibrozil isochlortetracycline, penicillin G sodium, cefotaxime, spectinomycin, roxithromycin, erythromycin, clarithromycin, thiamphenicol, chloramphenicol, paracetamol, naproxen, ibuprofen, ketoprofen, diclofenac acid, carbamazepine, diltiazem, diphenhydramine	0.2 (d.w)	Freeze-dried, ground into powder, mixed	Sonication (ACN/H_2_O)	SPE (HLB cartridges)	UHPLC-MS/MS	43–127	0.01–1.9	[39]
Zebra mussels (*Dreissena polymorpha*)	Nicotine, haloperidol, pyremethamine	0.14–0.2 (d.w)	Gut clearance, frozen, shelled, cryo-storage	USE (ACN, MeOH, H_2_O), vortex, USE	SPE (No data)	LC-HRMS/MS	70–130	0.05–5.70 *	[59]
Mussels (*Mytilus galloprovincialis*, *Mytilus edulis*)	Salicylic acid, clofibric acid, ketoprofen, naproxen, bezafibrate, diclofenac, ibuprofen	1 (d.w)	Lyophilized, homogenized	PLE (Ottawa sand, ultrapure water)	SPE (Oasis MAX cartridges)	LC-MS/MS	61–90	2–50	[66]
Clams (*Ruditapes decussatus*, *ruditapes philippinarum*)	Acetaminophen, clofibric acid, atenolol, bezafibrate, carbamazepine, cortisone, diclofenac, erythromycin, fluoxetine, ibuprofen, naproxen, propanolol, sulfadiazine, sulfapyridine, caffeine, sulfamethoxazole, testosterone, gestodene, metoprolol, diethylsilbestrol, estradiol, estriol, estrone, 17α-ethinylestradiol	1 (w.w)	Depurated, frozen at −20 °C, homogenized before analysis	Manual shaking (ACN)	QuEChERs (Hexane)	LC-MS/MS	35.2–118	0.35–5.86	[67]
Mussel (*Anodonta*), Snail (*Bellamya sp.*), Bivalve (*Corbiculidae*)	Roxithromycin, erythromycin, ofloxacin, norfloxacin, ciprofloxacin, tetracycline, chloramphenicol, sulfamerazine and sulfadiazine, sulfamethoxazole, ibuprofen, diclofenac, naproxen and indomethacin, clofibric acid, gemfibrozil and bezafibrate, 17β-estradiol and 17α-ethynylestradiol, propranolol, carbamazepine, ketoconazole, sertraline	0.5 (d.w)	Freeze-dried, ground, stored at −20 °C	PLE (MeOH/ acetone)	SPE (HLB cartridges)	LC-MS/MS	68–116	0.01–1.12	[56]
Asian clam (*Corbicula fluminea*)	Sulfachlorpyridazine, sulfadiazine, sulfadoxine, sulfamerazine, sulfadimethoxine, sulfamethazine, sulfamethoxazole, sulfamonomethoxine, sulfapyridine, sulfaquinoxaline, sulfisoxazole, sulfathiazole, trimethoprim, chlortetracycline, doxycycline, oxytetracycline, tetracycline, ciprofloxacin, danofloxacin, difloxacin, enrofloxacin, fleroxacin, lomefloxacin, marbofloxacin, norfloxacin, ofloxacin, pefloxacin, sarafloxacin, azithromycin, clarithromycin, leucomycin, oleandomycin, roxithromycin, tylosin, salinomycin, monensin, florfenicol, chloramphenicol	1 (d.w)	Washed (water), homogenized, freeze-dried, stored at −20 °C	USE (AcONa buffer/ MeOH)	SPE (SAX/PSA-HLB tandem cartridges)	RRLC-MS/MS	47.9–136.7	0.01–1.99	[38]
Mussel (*Anodonta woodiana*)	Roxithromycin, erythromycin, ofloxacin, norfloxaxin, ciprofloxacin, tetracycline, sulfamethoxazole, sulfadiazine, sulfaquinoxaline, ibuprofen, diclofenac, naproxen, bezafibrate, propranolol, ketoconazole, carbamazepine, caffeine, fluoxetine, norfluoxetine, citalopram, paroxetine, sertraline, venlafaxine, duloxetine, bupropion, amitriptyline, fluvoxamine, trihexylphenidyl, clozapine, quetiapine, aripiprazole, chlorpromazine	0.5 (d.w)	Freeze-dried, homogenized, stored at −80 °C	PLE (MeOH/ acetone)	SPE (HLB cartridges)	UHPLC-MS/MS	66–128	0.07–1.67	[27]
Clam (*Ruditapes decussatus*), Cockle (*Cerastodema glaucum*), Noble pen shell (*Pinna nobilis*), Sea snail (*Murex trunculus*)	Diclofenac, codeine, carbamazepine, citalopram, diazepam, lorazepam, atenolol, sotalol, propanolol, nadolol, carazolol, hydrochlorothiazide, clopidogrel, salbutamol, levamisole	1 (d.w)	Freeze-dried, milled	PLE (50 °C)	GPC, HPLC-DAD	UHPLC-MS/MS	<20–151.9	0.0004–6	[48]
Pen shell (*Atrina pectinate Linnaeus*), Asian hard clam (*Meretrix lusoria*), Magallana rivularis (*Crassostrea rivvularis Gould*).	Sulfadiazine, sulfadimethoxine, sulfadoxine, sulfamerazine, sulfameter, sulfamethazine, sulfamethoxazole, sulfamonomethoxine, sulfapyridine, sulfaquinoxaline, sulfathiazole, sulfisoxazole, trimethoprim, chlortetracycline, doxycycline, methacycline, oxytetracycline, tetracycline, ciprofloxacin, danofloxacin, difloxacin, enrofloxacin, fleroxacin, lomefloxacin, marbofloxacin, norfloxacin, ofloxacin, pefloxacin, clarythromycin, erythromycin-H_2_O, leucomycin, roxithromycin, oleandomycin	2 muscle (w.w)	Frozen, muscle dissected	USE (MeOH/ H_2_O 0.1 mol L^−1^ acetic acid)	SPE Cartridges (SAX/PSA and HLB cartridges)	LC-MS/MS	50–150	0.05–9.06	[42]
Clam (*Anadara ferruginea*)	Atenolol, metoprolol, venlafaxine, chloramphenicol	2 (d.w)	Washed (water), dissected, homogenized, freeze-dried, stored at −50 °C	USE (MeOH/ H_2_O)	SPE (MCX cartridges)	LC-MS/MS	68–96	0.05–0.25	[44]
**Gastropods**									
**Specie**	**Pharmaceuticals**	**Type and Amount of Sample (g)**	**Pre-Treatment**	**Treatment**		**Analysis**	**Analytical Features**		**Ref.**
				**Extraction** **Technique**	**Clean-Up**		**Recovery (%)**	**LOD** **(ng g^−1^)**	
Snail (*Bellamya aeruginosa*)	Sulfachlorpyridazine, sulfadiazine, sulfadoxine, sulfamerazine, sulfadimethoxine, sulfamethazine, sulfamethoxazole, sulfapyridine, sulfamonomethoxine, sulfaquinoxaline, sulfisoxazole, sulfathiazole, trimethoprim, chlortetracycline, doxycycline, oxytetracycline, tetracycline, ciprofloxacin, danofloxacin, difloxacin, enrofloxacin, fleroxacin, lomefloxacin, marbofloxacin, norfloxacin, ofloxacin, pefloxacin, sarafloxacin, azithromycin, leucomycin, clarithromycin, oleandomycin, roxithromycin, tylosin, salinomycin, monensin, florfenicol, chloramphenicol	1 soft tissues (d.w)	Washed (water), homogenized, freeze-dried, stored at −20 °C	USE (AcONa buffer/ MeOH)	SPE (SAX/PSA−HLB tandem cartridges)	RRLC-MS/MS	47.9–136.7	0.01–1.99	[38]
Snail (*Bellamya aeruginosa*)	Roxithromycin, erythromycin, ofloxacin, norfloxaxin, ciprofloxacin, tetracycline, sulfamethoxazole, sulfadiazine, sulfaquinoxaline, ibuprofen, diclofenac, naproxen, bezafibrate, propranolol, ketoconazole, carbamazepine, caffeine, fluoxetine, norfluoxetine, citalopram, paroxetine, sertraline, venlafaxine, duloxetine, bupropion, amitriptyline, clozapine, fluvoxamine, trihexylphenidyl, quetiapine, aripiprazole, chlorpromazine	0.5 (d.w)	Freeze-dried, homogenized, stored at −80 °C	PLE (MeOH/ acetone)	SPE (HLB cartridges)	UHPLC-MS/MS	66–128	0.07–1.67	[27]
Conch (*Bufonaria perelegans*)	Sulfadiazine, sulfamerazine, sulfamethazine, trimethoprim, sulfamethoxazole, sulfathiazole, sulfapyridine, ciprofloxacin, norfloxacin, ofloxacin, flumequine, tetracycline, oxytetracycline, isochlortetracycline, penicillin G sodium, cefotaxime sodium, spectinomycin, roxithromycin, erythromycin- H_2_O, clarithromycin, thiamphenicol, chloramphenicol, paracetamol, naproxen, ibuprofen, ketoprofen, diclofenac acid, carbamazepine, diltiazem, diphenhydramine, gemfibrozil	0.2 (d.w)	Freeze-dried, ground into powder. The whole body was mixed	USE (ACN/H_2_O)	SPE (PRiME HLB cartridges)	UHPLC-MS/MS	43–127	0.01–1.9	[39]
Sea snail (*Murex trunculus*)	Diclofenac, codeine, carbamazepine, citalopram, diazepam, lorazepam, atenolol, sotalol, propanolol, nadolol, carazolol, hydrochlorothiazide, clopidogrel, salbutamol, levamisole	1 (d.w)	Freeze-dried and milled	PLE (MeOH)	GPC, HPLC-DAD	UHPLC-MS/MS	<20–151.9	0.0004–6	[48]
Snail (*B. tentaculata*)	Ethinylestradiol, acetaminophen, diclofenac	0.35 (d.w)	Freeze-dried, powered	USE (ACN/MeOH 1% acetic acid)	No data	HPLC-MS/MS	67	No data	[55]
River limpet (*Ancylus fluviatilis*)	Diclofenac, ibuprofen, 1-OH-ibuprofen, piroxicam, acridone, propyphenazone, sulfamethoxazole, diltiazem, verapamil, norverapamil, hydrochlorothiazide, bezafibrate, gemfibrozil, pravastatin, carbamazepine, 10,11-epoxy-CBZ, 2-OH-CBZ, citalopram, fluoxetine, paroxetine, venlafaxine, azaperone, dexamethasone, metoprolol, propanolol	0.1 (d.w)	Homogenized with a mortar, kept at 20 °C	USE (MeOH)	Protein Precipitation and Phospholipid Removal, PlateOSTRO™ plate	UHPLC-MS/MS	No data	No data	[54]
Turritella bacillum Murex trapa, Bufonaria rana (No data)	Atenolol, metoprolol, venlafaxine, chloramphenicol	2 (d.w)	Washed (water), dissected, homogenized, freeze-dried, stored at −50 °C	USE (MeOH/ H_2_O)	SPE (MCX cartridges)	LC-MS/MS	68–96	0.05–0.25	[44]
**(e)**									
**Specie**	**Pharmaceuticals**	**Type and Amount of Sample (g)**	**Pre-Treatment**	**Treatment**		**Analysis**	**Analytical Features**		**Ref.**
				**Extraction** **Technique**	**Clean-Up**		**Recovery (%)**	**LOD** **(ng g^−1^)**	
Octopus (*Octopus vulgaris*)	Atenolol, metoprolol, nadolol, propanolol, sotalol, salbutamol, diazepam, carbamazepine, 10,11-epoxycarbamazepine, 2-OH-carbamazepine, citalopram, venlafaxine, alprazolam, azaperol, azaperone, hydrochlorothiazide, codeine, phenazone, propyphenazone, piroxicam, ronidazole, dimetridazole, metronidazole, azithromycin, erythromycin	1 (d.w)	Freeze-dried	PLE (MeOH/ H_2_O)	GPC	UHPLC-MS/MS	No data	0.02–0.3	[40]
Sepia (*Sepia indica*), Octopus (*Octopus rugosus*), Octopus minor (*Polypus variabilis*), Urotheutis (*Loligo oshimai*)	Sulfamethazine, sulfapyridine, sulfathiazole, sulfanlamide, sulfadiazine, sulfadimethoxine, sulfamonomethoxin, sulfamerazine, sulfamethoxazole, norfloxacin, enoxacin, ofloxacin, ciprofloxacin, enrofloxacin, dehydrated erythromycin, clarithromycin, azithromycin, roxithromycin, florfenicol, chloramphenicol, trimethoprim, lincomycin	5 (d.w)	Washed (water), dissected, homogenized, stored at −20 °C	USE (ACN, citric acid)	SPE (SAX-HLB cartridges)	UHPLC-MS/MS	47.7–172.7	0.04–0.24	[13]
**(f)**									
**Specie**	**Pharmaceuticals**	**Type and Amount of Sample (g)**	**Pre-Treatment**	**Treatment**		**Analysis**	**Analytical Features**		**Ref.**
				**Extraction** **Technique**	**Clean-up**		**Recovery (%)**	**LOD** **(ng g^−1^)**	
Starfish (*Marthasterias glacialis*), Sea urchins (*parechinus angulosus*)	Phenytoin, lamivudine, acetaminophen, caffeine, sulfamethoxazole, diclofenac, carbamazepine	10 (d.w)	Rinsed, deshelled, dissected, freeze-dried	Soxhlet (MeOH/ Acetone)	SPE (HLB cartridges)	UHPLC-MS/MS	96.1–100.5	0.62–1.05 ng L^−1^	[37]
**(g)**									
**Specie**	**Pharmaceuticals**	**Type and Amount of Sample (g)**	**Pre-Treatment**	**Treatment**		**Analysis**	**Analytical Features**		**Ref.**
				**Extraction** **Technique**	**Clean-Up**		**Recovery (%)**	**LOD** **(ng g^−1^)**	
Barnacle (*Balanus perforatus*)	Atenolol, ranitidine, acetaminophen, caffeine, trimethoprim, atrazine, amitriptyline, carbamazepine, chloropheniramine malate, ciprofloxacin, diclofenac, fluoxetine, ibuprofen, metronidazole, sulfamethoxazole, warfarin, cephalexin.	1 (d.w)	Dried, ground, pooled, homogenized, freeze-dried	USE (0.1 M acetic acid, MeOH)	SPE (Oasis MCX)	HPLC-MS/MS	30–103	0.1–13 ng mL^−1^	[52]
Shrimp (*Caridea*), Brown crab (*Cancer pagurus*)	Diclofenac, diazepam, sotalol, carbamazepine, citalopram, venlafaxine, azithromycin, sulfamethoxazole	All edible meat (no data)	Pooled, homogenized by grinding, freeze-dried, −20 °C	PLE (MeOH/ H_2_O)	SPE (HLB cartridges)	UHPLC-MS/MS	No data	0.01–0.65	[64]
Crabs (*Calappa philargius)*, pen shell *Atrina pectinate Linnaeus*), shrimps (*Fenneropenaeus penicillatus*)	Sulfadiazine, sulfadimethoxine, sulfadoxine, sulfamerazine, sulfameter, sulfamethazine, sulfamethoxazole, sulfapyridine, sulfamonomethoxine, sulfaquinoxaline, sulfathiazole, sulfisoxazole, trimethoprim, chlortetracycline, doxycycline, methacycline, oxytetracycline, tetracycline, ciprofloxacin, danofloxacin, difloxacin, enrofloxacin, fleroxacin, lomefloxacin, marbofloxacin, norfloxacin, ofloxacin, pefloxacin, clarythromycin, erythromycin-H_2_O, leucomycin, roxithromycin, oleandomycin	2 muscle (w.w)	Frozen and muscle dissected	USE (MeOH/ H_2_O 0.1 mol L^−1^ acetic acid	SPE (SAX/PSA, HLB cartridges)	LC-MS/MS	50–150	0.05–9.06	[42]
No data	Ketoprofen, naproxen, flurbiprofen, diclofenac sodium, ibuprofen	5 muscle tissue (w.w)	Chopped into mince	USE (ACN)	SPE (CF@UiO−66 NH_2_)	UHPLC-PDA	95–116.99	0.12–3.50 ng mL^−1^	[68]
Goose Barnacle (*Pollicipes*,) Carb (*Necora púber*)	Atenolol, metoprolol, nadolol, propanolol, sotalol, salbutamol, diazepam, carbamazepine, 10,11-epoxycarbamazepine, 2-OH-carbamazepine, citalopram, venlafaxine, alprazolam, azaperone, azaperol, hydrochlorothiazide, codeine, phenazone, propyphenazone, piroxicam, ronidazole, dimetridazole, metronidazole, azithromycin, erythromycin	1 (d.w)	Freeze-dried	PLE (MeOH)	GPC	UHPLC-MS/MS	No data	0.03–0.09	[40]
Shrimp (*Palaemon serratus*)	Metronidazole, acetaminophen, amoxicillin, acetazolamide, sulfadiazine, atenolol, caffeine, ampicillin, trimethoprim, norfloxacin, ofloxacin, ciprofloxacin, tetracycline, phenazone, metoprolol, spiramycin, azithromycin, sulfamethoxazole, oxolinic acid, erythromycin A, piperacillin, tylosine, cyclophosphamide, carbamazepine, flumequine, oxazepam, clarithromycin, roxithromycin, lorazepam, losartan, nordiazepam, josamycin, ketoprofen, 19-norethind-rone, amiodarone, hydrochlorothiazide, acetylsalicylic acid, niflumic acid, diclofenac, ibuprofen, gemfibrozil	0.2 (d.w)	Separated abdomen muscle, freeze-dried	Mechanical shaking (MeOH 1% acetic acid)	No data	UHPLC-MS/MS	26–132	0.1–40.2 *	[69]
Freshwater amphipod (*Gammarus pulex*)	Propanolol hydrochloride, ketoprofen, diclofenac salt, bezafibrate, warfarin, flurbiprofen, indomethacin, ibuprofen sodium salt, meclofenamic acid sodium salt, gemfibrozil, atenolol, sulfamethoxazole, sulfamethazine, furosemide, carbamazepone, nimesulide, (+-metoprolol) (+) tartrate, cimetidine, ranitidine, antipyrin, temazepam, diazepam, fluoxetine, nifedipine, mefenamic acid, trimethoprim, caffeine, naproxen	0.1 (d.w)	Freeze-dried, pulverized	PuLE (ACN)	SPE (HLB cartridges)	LC-MS/MS	41–89	1–13	[70]
Green crab (*Carcinus maenas*)	Alprazolan, amoxicillin, atenolol, atorvastatin, azithromycin, bisoprolol, benzylpenicillin, bezafibrate, carbamazepine, carvedilol, cinoxacin, ciprofloxacin, ceftiofur, cephalexin, chlortetracycline, danofloxacin, diclofenac, doxicycline, enoxacin, enrofloxacin, epi-chlortetracycline, epi-tetracycline, erythromycin, epotetracycline, fenofibrate, flumequine, fluoxetine, furosemide, gabapentin, gemfibrozil, ibersartan, ibuprofen, indapamide, lorazepam, losartan, marbofloxacin, nalidixic acid, norfloxacin, nimesulide, ofloxacin, oxolinic acid, oxytetracycline, paracetamol, propanolol, sertraline, simvastatin, spiramycin, sulfachloropyridazine, sulfadiazine, sulfadimethoxine, sulfamethazine, sulfamethizole, sulfanilamide, sulfapyridine, sulfisomidine, sulfadoxine, sulfamethoxazole, sulfaquinoxaline, sulfathiazole, sulfisoxazole, tetracycline, tilmicosin, trimethoprim, venlafaxine, topiramate	2 (w.w)	Homogenized	Mechanical shaking (ACN, EDTA)	No data	UHPLC-MS/MS	79.2–109.5	0.59–4.11	[41]
Shrimps: White vannamei prawn, Indian prawn, kiddi shrimp (No data)	Amoxicillin, azithromycin, caffeine, carbamazepine, ciprofloxacin, clarithromycin, diclofenac, erythromycin, furosemide, ketoprofen, ibuprofen, naproxen, sulfamethoxazole, tetracycline	2 (w.w)	Abdomen muscle separated, cut into small parts, frozen at −20 °C	Mechanical shaking (ACN, 0.1 M EDTA, hexane)	No data	UHPLC-MS/MS	81.2–99.4	0.017–1.371	[46]
Shrimp (*Harpiosquilla harpax*), Crab (*Charybdis japonica*), Spear shrimp (*Parapenaeopsis hardwickii*), Giant tiger prawn (*Penaeus monodon*), Green mud crab (*Scylla paramamosain*), Prawn (*Trachypenaeus sedili*)	Sulfamethazine, sulfapyridine, sulfathiazole, sulfanilamide, sulfadiazine, sulfadimethoxine, sulfamonomethoxin, sulfamerazine, sulfamethoxazole, norfloxacin, enoxacin, ofloxacin, ciprofloxacin, enrofloxacin, dehydrated, erythromycin, clarithromycin, azithromycin, roxithromycin, florfenicol, chloramphenicol, trimethoprim, lincomycin	5 (d.w)	Washed (water), dissected, homogenized, stored at −20 °C	USE (ACN, citric acid)	SPE (SAX-HLB cartridges)	UHPLC-MS/MS	47.67–172.67	0.04–0.24	[13]
Mud prawun (*Meapenaeus ensis*), Smoothshell shrimp (*Parapenaeopsis tenella*), Three-spot swimming crab (*Portunus sanguinolentus*), Jinga shrimp (*Metapenaeus affinis*), Robber harpiosquillid mantis shrimp (*Harpiosquilla harpax*)	Atenolol, metoprolol, venlafaxine, chloramphenicol	2 (d.w)	Washed (water), dissected, homogenized, freeze-dried, stored at −50 °C	USE (MeOH/ H_2_O)	SPE (MCX cartridges)	LC-MS/MS	68–96	0.05–0.25	[44]
White shrimp (*Exopalaemon modestus*) Taihu shrimp (*Macrobranchium nipponense*)	Roxithromycin, erythromycin, ofloxacin, norfloxacin, ciprofloxacin, tetracycline, chloramphenicol, sulfamerazine, sulfadiazine, sulfamethoxazole, ibuprofen, diclofenac, naproxen and indomethacin, clofibric acid, gemfibrozil, bezafibrate, 17β-estradiol,17α-ethynylestradiol, propranolol, carbamazepine, ketoconazole, sertraline	0.5 (d.w)	Separated muscle of shrimp. Freeze-dried, ground and stored at −20 °C	PLE (MeOH/ acetone)	SPE (HLB cartridges)	LC-MS/MS	68–116	0.01–1.12	[56]
Water flea (*Gammarus pulex*)	Ethinylestradiol, acetaminophen, diclofenac	0.34 (d.w)	Freeze-dry, powered	USE (ACN/ MeOH 1% acetic acid)	No data	HPLC-MS/MS	67	No data	[55]
Shrimps (*Paranthura sp.*, *Macrobrachium nipponense*), Crab (*Eriocheir sinensis*)	Sulfachlorpyridazine, sulfadiazine, sulfadoxine, sulfamerazine, sulfadimethoxine, sulfamethazine, sulfamethoxazole, sulfamonomethoxine, sulfapyridine, sulfaquinoxaline, sulfisoxazole, sulfathiazole, trimethoprim, chlortetracycline, doxycycline, oxytetracycline, tetracycline, ciprofloxacin, danofloxacin, difloxacin, enrofloxacin, fleroxacin, lomefloxacin, marbofloxacin, norfloxacin, ofloxacin, pefloxacin, sarafloxacin, azithromycin, clarithromycin, leucomycin, oleandomycin, roxithromycin, tylosin, salinomycin, monensin, florfenicol, chloramphenicol	1 (d.w)	Washed (water), homogenized, freeze-dried, stored at −20 °C	USE (sodium acetate buffer/ MeOH)	SPE (SAX/PSA−HLB tandem cartridges)	RRLC-MS/MS	47.9–136.7	0.01–1.99	[38]
Shrimp (*Macrobranchium nipponense*)	Roxithromycin, erythromycin, ofloxacin, norfloxaxin, ciprofloxacin, tetracycline, sulfadiazine, sulfamethoxazole, sulfaquinoxaline, naproxen, ibuprofen, diclofenac, bezafibrate, propranolol, ketoconazole, carbamazepine, caffeine, fluoxetine, norfluoxetine, citalopram, paroxetine, sertraline, venlafaxine, duloxetine, bupropion, amitriptyline, fluvoxamine, trihexylphenidyl, clozapine, quetiapine, aripiprazole, chlorpromazine	0.5 (d.w)	Freeze-dried, homogenized, stored at −80 °C	PLE (MeOH/ acetone)	SPE (HLB cartridges)	UHPLC-MS/MS	66–128	0.07–1.67	[27]
Shrimps (No data)	Naproxen, methyltestosterone, 17α-hydroxyprogesterone caproate, progesterone	2 (w.w)	Ground, homogenized	Manual shaking (ACN 0.1% acetic acid)	No data	LC-MS/MS	68–117	1–2	[43]
Crabs: Spectacled box crab (*Calappa philargius*). Shrimps: Redtail shrimpredtail prawn (*Fenneropenaeus penicillatus*)	Sulfadiazine, sulfadimethoxine, sulfadoxine, sulfamerazine, sulfameter, sulfamethazine, sulfamethoxazole, sulfapyridine, sulfamonomethoxine, sulfaquinoxaline, sulfathiazole, sulfisoxazole, trimethoprim, chlortetracycline, doxycycline, methacycline, oxytetracycline, tetracycline, ciprofloxacin, danofloxacin, difloxacin, enrofloxacin, fleroxacin, lomefloxacin, marbofloxacin, norfloxacin, ofloxacin, pefloxacin, clarythromycin, erythromycin- H_2_O, leucomycin, roxithromycin, oleandomycin	2 (w.w)	Frozen and muscle dissected	USE (MeOH/ H_2_O, 0.1 M acetic acid)	SPE Cartridges (SAX/PSA, and HLB cartridges)	LC-MS/MS	50–150	0.05–9.06	[42]
**(h)**									
**Specie**	**Pharmaceuticals**	**Type and Amount of Sample (g)**	**Pre-Treatment**	**Treatment**		**Analysis**	**Analytical Features**		**Ref.**
				**Extraction** **Technique**	**Clean-Up**		**Recovery (%)**	**LOD (ng g^−1^)**	
Ragworm (*Hedyste diversicolor*)	Alprazolan, amoxicillin, atenolol, atorvastatin, azithromycin, bisoprolol, benzylpenicillin, bezafibrate, carbamazepine, carvedilol, cinoxacin, ciprofloxacin, ceftiofur, cephalexin, chlortetracycline, danofloxacin, diclofenac, doxicycline, enoxacin, enrofloxacin, fluoxetine, epi-chlortetracycline, epi-tetracycline, erythromycin, epotetracycline, fenofibrate, flumequine, furosemide, gabapentin, gemfibrozil, ibersartan, ibuprofen, indapamide, lorazepam, losartan, marbofloxacin, nalidixic acid, norfloxacin, nimesulide, ofloxacin, oxolinic acid, oxytetracycline, paracetamol, propanolol, sertraline, simvastatin, spiramycin, sulfachloropyridazine, sulfadiazine, sulfadimethoxine, sulfamethazine, sulfamethizole, sulfanilamide, sulfapyridine, sulfisomidine, sulfadoxine, sulfamethoxazole, sulfaquinoxaline, sulfathiazole, sulfisoxazole, tetraccline, tilmicosin, trimethoprim, tylosin venlafaxine, topiramate	2.0 (w.w)	Homogenized	Mechanical shaking (ACN, EDTA)	No data	UHPLC-MS/MS	79.2–109.5	0.59–4.11	[41]
Polychaetas (*Perinereis aibuhitensis*, *Notomastus latericeus*, *Sabella pavonina*). Insecta (*Chironomidae sp.*). Worm (*Limnodrilus hoffmeisteri*)	Sulfachlorpyridazine, sulfadiazine, sulfadoxine, sulfamerazine, sulfadimethoxine, sulfamethazine, sulfamethoxazole, sulfamonomethoxine, sulfapyridine, sulfaquinoxaline, sulfisoxazole, sulfathiazole, trimethoprim, chlortetracycline, doxycycline, oxytetracycline, tetracycline, ciprofloxacin, danofloxacin, difloxacin, enrofloxacin, fleroxacin, lomefloxacin, marbofloxacin, norfloxacin, ofloxacin, pefloxacin, sarafloxacin, azithromycin, clarithromycin, leucomycin, oleandomycin, roxithromycin, tylosin, salinomycin, monensin, florfenicol, chloramphenicol	1.0 (d.w)	Washed (water), homogenized, freeze-dried, stored at −20 °C	USE (sodium acetate buffer/ MeOH)	SPE (SAX/PSA−HLB tandem cartridges)	RRLC-MS/MS	47.9–136.7	0.01–1.99	[38]
Porifera: Sponge (*Cf. Hyrtios*)	Caffeine, fluoxetine, norfluoxetine	0.25 (d.w)	Squeezed, wrapped in aluminium foil, and freeze-dried	USE (acidified methanol, acetonitrile/ methanol, acetonitrile)	SPE (HLB)	UHPLC-MS	80	0.01–10	[71]
Insecta (*Hydropsyche sp.*, *Phagocata vitta*)	Diclofenac, ibuprofen, 1-OH-ibuprofen, piroxicam, propyphenazone, sulfamethoxazole, diltiazem, verapamil, norverapamil, hydrochlorothiazide, bezafibrate, gemfibrozil, pravastatin, carbamazepine, acridone, 10,11-epoxy-CBZ, 2-OH-CBZ, citalopram, fluoxetine, paroxetine, venlafaxine, dexamethasone, azaperone, metoprolol, propanolol	0.1 (d.w)	Homogenized with a mortar, kept at 20 °C	USE (MeOH)	Protein precipitation and phospholipid removal, PlateOSTRO™ plate	UHPLC-MS/MS	No data	No data	[54]
**(i)**									
**Specie**	**Pharmaceuticals**	**Type and Amount of Sample (g)**	**Pre-Treatment**	**Treatment**		**Analysis**	**Analytical Features**		**Ref.**
				**Extraction** **Technique**	**Clean-Up**		**Recovery (%)**	**LOD** **(ng g^−1^)**	
Surgeonfish (*Acanthurus xanthoperus*), Smallmouth catfish (*Ariopsis felis*), Bull fish (*Caranx caninus*), Milkfish (*Chanos chanos*), Yellowfin mojarra (*Gerres cinereus*), Elongated grunt (*Haemulopsis elongatus*), Silk snapper (*Lutjanus peru*), *White mullet* (*mugil curema*), California halibut (*Paralichthys californicus*), Bigscale goatfish (*Pseudupeneus grandisquamis*), Peruvian moonfish (*Selene peruvian*), Common snook (*Centropomus robalito*), Reef Lizardfish (*Synodus lacertinus*), Striped bonito (*Sarda orientalis*)	Diclofenac, ibuprofen, ketorolac, naproxen	25–30 (w.w)	Minced, homogenized	USE (No data)	No data	UHPLC-MS/MS	92–95	0.97–23.1	[72]
Black Crappie (*Pomoxis nigromaculatus*), Black Redhorse (*Moxostoma duquesni*), Bluegill (*Lepomis macrochirus*), Common Carp (*Cyprinus carpio*), Flathead Catfish (*Pylodictis olivaris*), Freshwater Drum (*Aplodinotus grunniens*), Gizzard Shad (*Dorosoma cepedianum*), Golden Redhorse (*Moxostoma erythrurum*), Hybrid White x Striped Bass (*Morone chrysops x Morone saxatilis*), Largemouth Bass (*Micropterus salmoides*), Mooneye (*Hiodontidae*), Nothern Hogsucker (*Hypentelium nigricans*), Quillback Carpsucker (*Carpiodes cyprinus*), River Carpsucker (*Carpiodes carpio*), Sauger (*Sander canadensis*), Saugeye (*Sander canadensis x Sander vitreus*), Silver Redhorse (*Moxostoma anisurum*), Smallmouth Bass (*Micropetrus dolomieu*), Smallmouth Buffalo (*Ictiobus bubalus*), Smallmouth Redhorse (*Moxostoma breviceps*), Spotted Sucker (*Minytrema melanops*), White Bass (*Morone chrysops*), White Crappie (*Pomoxis annularis*)	Tylosin, lincomycin, furazolidone, sulfadimethoxine, sulfamethazine, sulfamethoxazole, sulfanilamide, cotinine, carbamazepine, acetaminophen, thiamphenicol, florfenicol, chloramphenicol, caffeine, trimethoprim, azithromycin, triclosan erythrohydrobupropion	0.5 (w.w)	Homogenized	QuEChERs (ACN/H20 1% acetic acid, MgSO4, AcONa)	d-SPE: QuEChERs (MgSO4, PSA, C18)	UHPLC-MS/MS	67–148	0.2–2.6	[73]
Perch (*Perca fluviatilis*), Flounder (*Platichthys flesus*), Turbot (*Scophthalmus maximus*), Plaice (*Pleuronectes platessa*), Cod (*Gadus morhua callarias*), Bream (*Abramis brama*), Crucian (*Carassius carassius*)	Bisoprolol, carbamazepine, clarithromycin, erythromycin, fluoxetine, metronidazole, ofloxacin, promazine, sulfadimethoxine, thiabenzadole, tianeptine, acebutolol, 1-Naphthoxyacetic acid, amitriptyline, amlodipine, atenolol, azithromycin, bosentan, cefotaxime, chlorpromazine, chlortetracycline, clindamycin, clomipramine, codeine, desipramine, dextromethorphan, diclofenac, diltiazem, doxepin, drotaverine, duloxetine, enalapril, escitalopram, fenofibrate, fleroxacin, fluconazole, fluvoxamine, guaifenesin, imipramine, labetalol, losartan, levofloxacin, lincomycin, lomefloxacin, lovastatin, maprotiline, mebendazole, metformin, methoxyverapamil, metoprolol, mianserin, mirtazapine, moclobemide, morantel, mycophenolic acid, nalidixic acid, nifedipine, norfloxacin, nortriptyline, omeprazole, opipramol, oxymetazoline, oxytetracycline, pantoprazole, paroxetine, pefloxacin, piperacillin, propafenone, propanolol, protriptyline, pseudophedrine, quinapril, ramipril, ranitidine, roxithromycin, salbutamol, sotalol, sertraline, sulfadiazine, sulfamethazine, sulfamethoxazole, sulfanilamide, sulfathiazole, telmisartan, tetracycline, tiamulin, tianeptine, tolperisone, trazodone, trimethoprim, tylosin, valsartan, verapamil, xylometazoline	0.05 (w.w)	Homogenized	Mechanical shaking (ACN 0.1% formic acid), frozen, centrifuged. Added ammonium acetate and stirred	d-SPE: C18 sorbent	LC-QTRAP	No data	0.01–0.88	[74]
Rainbow trout (*Oncorhynchus mykis*)	Citalopram	Brain tissue (no data)	Brain separated	TissueLyser II at 30 Hz for 10 min. (ACN:i-propanol 3:1 with 0.1% formic acid)	No data	LDTD- HRPS	97–108	0.39	[75]
Bream (*no data*)	Bezafibrate, carbamazepine, 2-hydroxicarbamazepine, 10,11-dihydroxy-10,11-dihydrocarbamazepine, cetirizine, citalopram, desmehylcitalopram, clopidogrel, diclofenac, diphenhydraine, fexodenadine, fluconazole, norfluoxetine, furosemide, hydrochlorothiazide, metoprolol, oxazepam, primidone, sertraline, sulfamethoxazole, trimethoprim, N-acetylsulfamethoxazole, telmisartan, tramadol, valsartan, venlafaxine, O-desmethylvenlafaxine.	0.05 (fish liver), 0.1 for (fish fillet) (d.w)	Homogenized, lyophilized	Cell disruption (4 m/s for 40 s)	d-SPE: Silica gel	LC-MS/MS	70–130	0.05–5.5 ng mL^−1^ *	[76]
Gilthead sea bream (*Sparus aurata*), Sea bass (*Dicentrarchus labrax*)	Ciprofloxacin, danofloxacin, difloxacin, enrofloxacin, flumequine, marbofloxacin, norfloxacin, ofloxacin, oxolinic acid, sarafloxacin, chlortetracycline, doxycycline, minocycline, oxytetracycline, tetracycline, cefaclor, cefadroxil, cefalexin, cefapirin, ceftiofur, cefazolinamoxicillin, ampicillin, cloxacillin, dicloxacillin, oxacillin, penicillin G, penicillin V, azithromycin, clarithromycin, erythromycin- H_2_O, tiamulin, tilmicosin, dapsone, sulfachlorpyridazine, sulfaclozine, sulfadiazine, sulfadoxine, sulfadimethoxine, sulfadimidine, sulfaguanidine, sulfameter, sulfamerazine, sulfamethizole, sulfamethoxazole, sulfamethoxypuridazine, sulfaonomethoxine, sulfamoxole, sulfapyridine, sulfaquinoxaline, sulfathiazole, sulfisoxazole, carbadox, olaquindox, florfenicol, thiampenicol, baquiloprin, trimthoprim, lincomycin, novobiocin, rifaximin, albendazole, albendazole oxide, albendazole sulfone, febantel, dimetridazole, fenbendazole, flubendazole, morantel, levamisole, mebendazole, metronidazole, oxfendazole, piperazine, ronidazole, ternidazole, thiabenzadole, triclabendazole, arprinocid, clopidol, decoquinate, diaveridine, ethopabate, halofuginone, imidocarb, lasalocid, monensin, narasin, nigericin, robenidine, salinomycin, 5-hydroxyflunixin, aceclofenac, diclofenac, flunixin, ketoprofen, mefenamic acid, naproxen, meloxicam, niflumic acid, phenylbuntazone, tolfenamic acid, vedaprofen, cimaterol, clenbuterol, clenpenterol, mabuterol, ractopamine, salbutamol, terbutaline, betamethasone, cortisol, cortison, dexamethazone, methyl-thiouracil, methylprednisolone, progesteron, phenyl- thiouracil, propyl-thiouracil, ambroxol, atenolol, atorvastatin, caffeine, carbamazepine, cimetidine, gemfibrozil, haloperidol, indapamide, metformin, metoprolol, paracetamol, propranolol, ranitidine, simvastatin, theophyline, tramadol, triamterene, valsartan, bromhexine, chlorpromazine, colchicine, melamine, coumaphos	1.0 (w.w)	Homogenized, stored at −20 °C	Ultrasonic bath (H_2_O containing 0.1% formic acid, 0.1% EDTA (*w*/*v*), MeOH, ACN). Precipitation of lipids and proteins	Hexane and further low temperature	UHPLC-MS/MS	No data	20–200	[77]
No data	Chloramphenicol, thiamphenicol, tinidazole, metronidazole, malachite green, crystal violet	2.0 (d.w)	Cleaned, scaled and muscle tissue was taken. Homogenised, blotted dried, freeze at −20 °C	MAE (ACN)	SPE (Activated neutral alumina column), USE (ACN) and DLLME (H_2_O, CH_2_Cl_2_, ACN)	UHPLC-MS/MS	>87	4.54–101.3 pg kg^−1^	[78]
Sea bream (*Sparus aurata*)	Erythromycin, N-acetyl sulfamethoxazole, sulfadiazine, sulfamethazine, sulfamethizole, sulfamethoxazole, sulfamethoxypyridazine, sulfapyridine, sulfaquinoxaline, sulfathiazole, trimethoprim, caffeine, paracetamol, phenazone, carbamazepine, carbamazepine-10,11- epoxide, citalopram, fluoxetine, N desmethyl sertraline, norfluoxetine, O desmethyl venlafaxine, sertraline, venlafaxine	1.0 (w.w)	Filleted	QuEChERs (ACN, MgSO_4_, NaCl)	d-SPE: Z-Sep+	UHPLC-MS/-MS	62–107	0.5–19 *	[79]
Sonek (*Thyrsites atun*), Bonito (*Sarda orientalis*), Panga (*Pachymetopon blochii*), Hottentot (*Pterogymnus laniarius*)	Acetaminophen, caffeine, diclofenac, lamivudine, sulfamethoxazole, carbamazepine	10 (d.w)	Dissection of different parts (fillet, gills, liver and intestine), freeze-dried and ground	Soxhlet (MeOH/ Acetone)	SPE (HLB cartridges)	UHPLC-MS/MS	69.2–107.5	0.010–0.036	[49]
Sabalo (*Prochilodus lineatus*), Boga (*Megaleporinus obtusidens*), Dorado (*Salminus brasiliensis*)	Atenolol, carazolol, metoprolol, nadolol, propanolol, sotalol, diazepam, lorazepam, carbamazepine, 10,11-epoxycarbamazepine, 2-hydroxycarbamazepine, venlafaxine, clopidogrel, salbutamol, codeine, diclofenac, hydrochlorothiazide	1.0 (d.w)	Pooled, homogenized	PLE (MeOH)	GPC	UHPLC-MS/MS	26–115	0.028–2.7	[53]
Carps (*Carassius*), Japanese medakas (*Oryzias latipes*), Mosquitofish (*Gambussia affinis*)	Diclofenac, indomethacin, mefenamic acid, ibuprofen, bezafibrate, fenofibric acid, clofibric acid, gemfibrozil, diltiazem, amlodipine, propanolol, carvedilol, losartan, telmisartan, irbesartan, valsartan, rebamipide, cetirizine, diphenhydramine, chlorpheniramine, fexofenadine, epinastine, warfarin, tramadol, O-desmethyl tramadol, N-desmethyl tramadol, sertraline, norsertraline, fluoxetine, norfluoxetine, paroxetine, citalopram, venlafaxine, haloperidol, risperidone, quetiapine, chlorpromazine, aripiprazole, zotepine, phentyon, carbamazepine, clonazepam, diazepam, zolpidem, nitrazepam, oxazepam, flunitrazepam, lorazepam, alprazolam, etizolam, sulfapyridine, sulfamerazine, sulfisozole, sulfamethizole, sulfamethazine, sulfamonomethoxine, sulfamethoxazole, sulfadimethoxine, trimethoprim, lincomycin, fluconazole erythromycin, clarithromycin, rixothromycin, florfenicol	200 µL plasma (*Carassius carassius*) and 0.1 g whole-body tissue (rest)	Homogenized	USE (MeOH/ ACN, and acetic acid- ammonium acetate buffer)	SPE (HybridSPE^®^-Phospholipid cartridge)	LC-MS/MS	70–120	0.0077–0.93 ng mL^−1^	[50]
European eel (*Anguilla anguilla*)	Acetaminophen, atenolol, caffeine, diclofenac, etoricoxib, ibuprofen, naproxen, salicylic acid, triclosan, vildagliptin)	1.0 pool (w.w)	Pooled, chopped, and homogenized	d-SPE: QuEChERs (ACN, MgSO4, NaCl, DCS and TCD)	d-SPE: EMR-Lipid (MgSO_4_ and NaCl)	UHPLC-MS/MS	70–120	1.4–12	[51]
Rainbow trout (*Oncorhynchus mykiss*)	Enrofloxacin, norfloxacin, ciprofloxacin	5.0 muscle (w.w)	Boned	SPE (0.1 M K_2_HPO_4_ (pH = 6.5))	SPE (Strata XC cartridges)	LC-MS/MS	91.1–108.9	3.3–3.6	[80]
Nile Tilapia (*Oreochromis niloticus*), Milk fish (*chanos chanos*), Common silver biddy (*gerres oyena*), Golden snapper (*lutjanus johni*), Emperor fish (*ethrinus nebulosus*)	Atenolol, ranitidine, acetaminophen, caffeine, trimethoprim, atrazine, amitriptyline, carbamazepine, chloropheniramine malate, ciprofloxacin, diclofenac, fluoxetine, ibuprofen, metronidazole, sulfamethoxazole, warfarin, cephalexin.	1.0 (d.w)	Filleted and cut into small sections and lyophilized. Pooled and homogenized	USE (0.1 M aqueous acetic acid/MeOH and NH_4_OH 0.1 M)	SPE (Oasis MCX cartridges)	HPLC-MS/MS	30–103	0.1−13 ng mL^−1^	[52]
Mackerel (*Scomber scombrus*), tuna (*Thunnus thynnus*), cod (*Gadus morhua*), perch (*Perca fluviatilis*), Pangas catfish (*Pangasius pangasius*), sole (*Solea solea*), seabream (*Sparus aurata*), plaice (*Pleuronectes platessa*), salmon (*Salmonidae*)	Diclofenac, diazepam, sotalol, carbamazepine, citalopram, venlafaxine, azithromycin, sulfamethoxazole	Fillet (no data)	Pooled, homogenized by grinding, freeze-dried, kept at −20 °C	PLE (MeOH)	GPC	UHPLC-MS/MS	No data	0.01–0.65	[64]
Rusell’s snapper (*Lutjanus ruselli*), Saddle tailed sea perch (*Lutjanus erythopterus*), Silverfish (*Trachinotus ovatus*)	Sulfadiazine, sulfadimethoxine, sulfadoxine, sulfamerazine, sulfameter, sulfamethazine, sulfamethoxazole, sulfapyridine, sulfamonomethoxine, sulfaquinoxaline, sulfathiazole, sulfisoxazole, trimethoprim, chlortetracycline, doxycycline, methacycline, oxytetracycline, tetracycline, ciprofloxacin, danofloxacin, difloxacin, enrofloxacin, fleroxacin, lomefloxacin, marbofloxacin, norfloxacin, ofloxacin, pefloxacin, clarythromycin, erythromycin, leucomycin, roxithromycin, oleandomycin	2 (w.w)	Frozen, muscle dissected	USE: MeOH/H_2_O, 0.1 M acetic acid	SPE (SAX/PSA, HLB cartridges)	LC-MS/MS	50–150	0.05–9.06	[42]
No data	Ketoprofen, naproxen, flurbiprofen, diclofenac, ibuprofen	5 (w.w)	Chopped into mince	USE (ACN)	SPE CF@UiO-66 NH_2_	UHPLC- PDA	95–116.99	0.12–3.50 ng mL^−1^	[68]
European pilchardus (*Sardina pilchardus*)	Atenolol, metoprolol, nadolol, propanolol, sotalol, salbutamol, diazepam, carbamazepine, 10,11-epoxycarbamazepine, 2-OH-carbamazepine, citalopram, venlafaxine, alprazolam, azaperone, azaperol, hydrochlorothiazide, codeine, phenazone, propyphenazone, piroxicam, ronidazole, dimetridazole, metronidazole, azithromycin, erythromycin	1 (d.w)	Freeze-dried	PLE (MeOH, 4 extraction cycles)	GPC	UHPLC-MS/MS	No data	0.1–0.6	[40]
Hake (*Merluccius merluccius*), Red mullet (*Mullus surmuletus*), Sole (*Solea solea*)	Metronidazole, acetaminophen, amoxicillin, acetazolamide, sulfadiazine, atenolol, caffeine, ampicillin, trimethoprim, norfloxacin, ofloxacin, ciprofloxacin, tetracycline, phenazone, metoprolol, spiramycin, azithromycin, sulfamethoxazole, oxolinic acid, erythromycin A, piperacillin, tylosine, cyclophosphamide, carbamazepine, flumequine, oxazepam, clarithromycin, roxithromycin, lorazepam, losartan, nordiazepam, josamycin, ketoprofen, 19-norethind-rone, amiodarone, hydrochlorothiazide, acetylsalicylic acid, niflumic acid, diclofenac, ibuprofen, gemfibrozil	0.2 (d.w)	Separated white dorsal muscle, freeze-dried.	Mechanical shaking (MeOH, 1% acetic acid)	No data	UHPLC-MS/MS	28–188	0.1–40.2 *	[69]
Sea bream (*Sparus aurata*)	Trimethoprim, ciprofloxacin, norfloxacin, sulfadiazine, sulfamethoxazole, amitriptyline, clomipramine, imipramine, nortriptyline, eprosartan, irbesartan, losartan, telmisartan, valsartan, propanolol, acetaminophen, diclofenac, ketoprofen, bezafibrate, clofibric acid, carbamazepine, phenytoin	0.5 fish muscle and liver; 0.1 fish gills and brain (d.w)	Freeze-dried, ground, homogenized	FUSLE (MeOH/ H_2_O)	SPE (HLB cartridges)	LC-MS/MS	71–126	4–48	[79]
Mullet (*Mugil* spp., *Mugil curema*)*,* Snook (*Centropomus* spp.)	Bezafibrate, carbamazepine, chloramphenicol, diclofenac, 4′-Hydroxydiclofenac, furosemide, gemfibrozil, ibuprofen, indapamide, ketoprofen, naproxen, simvastatin	0.5 (d.w)	Dissection to obtain the morphometric measures, freeze-dried	QuEChERs (ACN, formic acid, NH_4_Cl)	d-SPE: QuEChERs (MgSO_4_, Z-Sep)	HPLC-MS/MS	70–133	0.004–2.16	[43]
Golden grey mullet (*Liza aurata*), Black goby (*Gobius niger*)	Diclofenac, codeine, carbamazepine, citalopram, diazepam, lorazepam, atenolol, sotalol, propanolol, nadolol, carazolol, hydrochlorothiazide, clopidogrel, salbutamol, levamisole	1.0 golden grey mullet muscle and black goby; 0.5 liver golden grey mullet (d.w)	Freeze-dried, milled	PLE (MeOH)	GPC	UHPLC-MS/MS	<20–200	0.02–6.6	[48]
Yellow grouper (*Epinephelus awoara*), Orbfish (*Ephippus orbis*), Topmouth culter (*Culter alburnus*)	Sulfadiazine, sulfamerazine, sulfamethazine, trimethoprim, sulfamethoxazole, sulfathiazole, sulfapyridine, ciprofloxacin, norfloxacin, ofloxacin, flumequine, tetracycline, penicillin G sodium, oxytetracycline, isochlortetracycline, cefotaxime sodium, spectinomycin, roxithromycin, erythromycin- H_2_O, clarithromycin, thiamphenicol, chloramphenicol, paracetamol, naproxen, ibuprofen, ketoprofen, diclofenac acid, diltiazem, carbamazepine, diphenhydramine, gemfibrozil	0.2 (d.w)	Freeze-dried, ground into powder. Separation of back muscles and abdominal muscles	USE (ACN/H_2_O)	SPE (PRiME HLB cartridges)	UHPLC-MS/MS	43–127	0.01–1.9	[39]
Senegal seabram (*Diplodus bellottii*), European sea bass (*Dicentrarchus labrax*), Meagre (*Argyrosomus regius*), Lusitanian toadfish (*Halobatrachus didactylus*)	Alprazolan, amoxicillin, atenolol, atorvastatin, azithromycin, bisoprolol, benzylpenicillin, bezafibrate, carbamazepine, carvedilol, cinoxacin, ciprofloxacin, ceftiofur, cephalexin, chlortetracycline, danofloxacin, diclofenac, doxicycline, enoxacin, enrofloxacin, epi-chlortetracycline, epi-tetracycline, erythromycin, epotetracycline, fenofibrate, flumequine, fluoxetine, furosemide, gabapentin, gemfibrozil, ibersartan, ibuprofen, indapamide, lorazepam, losartan, marbofloxacin, nalidixic acid, norfloxacin, nimesulide, ofloxacin, oxolinic acid, oxytetracycline, paracetamol, propanolol, sertraline, simvastatin, spiramycin, sulfachloropyridazine, sulfadoxine, sulfadiazine, sulfadimethoxine, sulfamethazine, sulfamethizole, sulfanilamide, sulfapyridine, sulfisomidine, sulfamethoxazole, sulfaquinoxaline, sulfathiazole, sulfisoxazole, tetracycline, tilmicosin, trimethoprim, tylosin venlafaxine, topiramate	2 (w.w)	Homogenized	Mechanical shaking (ACN, EDTA)	No data	UHPLC-MS/MS	79.2–109.5	0.59–4.11	[41]
White bream, roach, bleak, perch, asp, pike, pikeperch (No data)	Nicotine, haloperidol, pyremethamine	0.14–0.2 (d.w)	Dissected into fillet and carcass, frozen	USE (ACN, MeOH, H_2_O)	SPE (No data)	LC-HRMS/MS	70–130	0.05–5.7 ^*^	[59]
Atlantic salmon, Atlantic sea wolf, rainbow trout, Atlantic cod (No data)	Amoxicillin, azithromycin, caffeine, carbamazepine, ciprofloxacin, clarithromycin, diclofenac, erythromycin, furosemide, ketoprofen, ibuprofen, naproxen, sulfamethoxazole, tetracycline	2 (w.w)	Dorsal muscle separated, cut into small parts, frozen at −20 °C	Mechanical shaking (ACN, 0.1 M EDTA, hexane)	No data	UHPLC-MS/MS	81.2–99.4	0.017–1.371	[46]
Flatfish (No data)	Albendazole, 2-amino albendazole sulfone, albendazole sulfone, albendazole sulfoxide, febantel, fenbendazole, flubendazole, 2-amino flubendazole, oxfendazole, oxfendazole sulfone, oxibendazole, cefapirin, desacetylcefapirin, cefazoline, cefoperazone, halofuginone, azithromycin, tildipirosin, dimetridazole, ipronidazole, ipronidazole-OH, metronidazole, metronidazole-OH, tinidazole, ronidazole, dicloxacillin, nafcillin, oxacillin, penicillin V, 2-hydroxymethyl^−1^-methyl-5-nitromidazole, 4-methylaminoantipyrine, sarafloxacin, orbifloxacin, carbadox, quinoxaline-2-carboxylic acid, olaquindox, 3-methylquinoxaline-2-carboxilic acid, dapsone, N-acetyl dapsone, sulfapyridine, arprinocid, azaperol, azaperon, carazolol, caffeine, clenbuterol, clochicine, diphehydramine, flunixin, imidocarb, isometamidium, ketoprofen, loperamide, metoclopramide, nitroxynil, phenacetin, ractopamine, scopolamine, triamcinolone, valnemuline	2 (w.w)	Homogenized, stored at −20 °C	Mechanical shaking (Water/ ACN)	d-SPE: C18	UHPLC-MS/MS	73.2–115	0.5–5 *	[81]
Goldsilk seabream (*Acanthopagrus berda*), Indo-Malaysian barracuda (*Sphyraena jello*), Pale-edged stingray (*Dasyatis zugei*), Japanese scaled sardine (*Sardinella zunasi*), Yellow seabream (*Acanthopagrus latus*), Spotted scat (*Scatophagus argus*), Dotted gizzard shad (*Konosirus punctatus*), Porgies (*Acanthopagrus schleg*), Grey large-eye bream (*Gymnocranius griseus*), Pompano (*Trachinotus ovaus*), Saddleback silver (*Gerres limbatus*), Asian seabasses (*Lateolabrax maculatus*), Silver sea bream (*Rhabdosargus sarba*), Rough flathead (*Grammoplites scaber*), Bloch’s gizzard shad (*Nematalosa nasus*), Gangetic anchovy (*Thryssa mysiax*), Japanese goatfosh (*Upeaneus japonicus*), Genus (*Johnius fasciatus*)	Sulfamethazine, sulfapyridine, sulfathiazole, sulfanilamide, sulfadiazine, sulfadimethoxine, sulfamonomethoxin, sulfamerazine, sulfamethoxazole, norfloxacin, enoxacin, ofloxacin, ciprofloxacin, enrofloxacin, dehydrated erythromycin, clarithromycin, azithromycin, roxithromycin, florfenicol, chloramphenicol, trimethoprim, lincomycin	5 (d.w)	Washed (water), dissected, homogenized, stored at −20 °C	USE (ACN, citric acid)	SPE (SAX-HLB cartridges)	UHPLC-MS/MS	47.67–172.67	0.04–0.24	[13]
Eel, flatfish (No data)	Naproxen, methyltestosterone, 17α-hydroxyprogesterone caproate, progesterone	2 (w.w)	Ground, homogenized	Manual shaking (ACN 0.1% acetic acid)	No data	LC-MS/MS	68–117	1–2	[43]
Silver carp (*Hypophthalmichthys molitrix*), Bighead carp (*Aristichthys nobilis*), Common carp (*Cyprinus carpio*), Goldfish (*Carassius auratus*), Common skygazer (*Cultrichthys erythropterus*), Topmouth culter (*Culter alburnus*), Japanese grenadier anchovy (*Coilia ectenes taihuensis*), Asian pencil halfbeak (*Hyporhamphus intermedius*), Clearhead icefish (*Protosalanx hyalocranius*), Common sawbelly (*Hemiculter leucisculus*), Bitterling (*Rhodeus sinensis*), River sand pond snakehead (*Odontobutis potamophila*), Yellow catfish (*Pelteobagrus fulvidraco*), Asian swamp eel (*Monopterus albus*)	Sulfachlorpyridazine, sulfadiazine, sulfadoxine, sulfamerazine, sulfadimethoxine, sulfamethazine, sulfamethoxazole, sulfamonomethoxine, sulfapyridine, sulfaquinoxaline, sulfisoxazole, sulfathiazole, trimethoprim, chlortetracycline, doxycycline, oxytetracycline, tetracycline, ciprofloxacin, danofloxacin, enrofloxacin, fleroxacin, difloxacin, lomefloxacin, marbofloxacin, norfloxacin, ofloxacin, pefloxacin, sarafloxacin, azithromycin, leucomycin, clarithromycin, oleandomycin, roxithromycin, tylosin, salinomycin, monensin, florfenicol, chloramphenicol	0.2 liver; 1.0 muscle (d.w)	Washed (water), dissected, homogenized, freeze-dried, stored at −20 °C	No data	No data	RRLC-MS/MS	37.6–135	0.01–1.99	[38]
Grass carp (*Ctenopharyngodon idellus*), Silver carp (*Hypophtha lmichthys molitrix*), Common carp (*Cyprinus carpio*), Crucian carp (*Carassius auratus*), Bighead carp (*Hypophthalmichthys nobilis*), Whitebait (*Reganisalanx brachyrostralis*), Yellow catfish (*Pelteobagrus fulvidraco*), Catfish (*Silurus asotus*), Loach (*Paramisgurnus dabryanus*)	Roxithromycin, erythromycin, ofloxacin, norfloxaxin, ciprofloxacin, tetracycline, sulfamethoxazole, ibuprofen, sulfaquinoxaline, sulfadiazine, diclofenac, naproxen, bezafibrate, propranolol, ketoconazole, carbamazepine, caffeine, fluoxetine, norfluoxetine, citalopram, paroxetine, sertraline, venlafaxine, duloxetine, bupropion, amitriptyline, fluvoxamine, trihexylphenidyl, clozapine, quetiapine, aripiprazole, chlorpromazine	0.5 (d.w)	Freeze-dried, homogenized, stored at −80 °C	PLE (MeOH/ acetone)	SPE (HLB cartridges)	UHPLC-MS/MS	66–128	0.07–1.67	[27]
Red bigeye (*Priacanthus macracanthus*), Horn dragonet (*Callionymus curvicornis*), White-spotted spinefoot (*Siganus canaliculatus*), Silver jewfish (*Pennahia argentata*), Burrowing goby (*Trypauchen vagina*), Threadfin porgy (*Evynnis cardinalis*), Palad (*Solea ovata*), Anchovy (*Thryssa kammalensis*), Bony fishes (*Johnius heterolepis*), Japanese flathead (*Inegocia japonica*), Shortnose ponyfish (*Leiognathus brevirostris*), Big head croaker (*Collichthys lucidus*), Goatee croaker (*Dendrophysa russelii*), Yellow croaker (*Larimichthys crocea*), Largehead hairtail (*Trichiurus lepturus*)	Atenolol, metoprolol, venlafaxine, chloramphenicol	2 (d.w)	Washed (water), dissected, homogenized, freeze-dried, stored at −50 °C	USE (MeOH/ H_2_O)	SPE (Oasis MCX cartridges)	LC-MS/MS	68–96	0.05–0.25	[44]
Silver carp (*Hypophtha lmichtyts molitrix*), Common carp (*Cyprinus carpio*), Crucian carp (*Carassius auratus*), Lake anchovy (*Coilia extenes*), whitebait (*Reganisalanx brachyrostralis*), Redfin culter (*Cultrichthys erythropterus*), Yellow catfish (*Pelteobagrus fulvidraco*)	Roxithromycin, erythromycin, ofloxacin, norfloxacin, ciprofloxacin, tetracycline, chloramphenicol, sulfamerazine and sulfadiazine, sulfamethoxazole, ibuprofen, diclofenac, naproxen, indomethacin, clofibric acid, gemfibrozil, bezafibrate, 17β-estradiol, 17α-ethynylestradiol, propranolol, carbamazepine, ketoconazole, sertraline	0.5 (d.w)	Separation of liver, brain, gills, and muscle. Freeze-dried, ground, stored at −20 °C	PLE (MeOH/ acetone)	SPE (HLB cartridges)	LC-MS/MS	68–116	0.01–1.12	[56]
Crucian carp (*Carcassius carcassius*)	Florfenicol, thiamphenicol, ofloxacin, pipemidic acid	No data	Liver, muscle, gill and bile separated, washed with 0.15 M KCl, stored at −20 °C	USE (0.1 M AcONa, MeOH)	SPE (SAX/PSA-HLB tandem cartridges)	LC-MS/MS	79.2–91.0	0.5–0.6	[82]

* Limit of quantification; ACN: acetonitrile; DAD: diode-Array detection; DLLME: dispersive liquid-liquid microextraction; DMSO: dimethyl sulfoxide; d-SPE: dispersive solid phase extraction; d.w.: dry weight; FUSLE: focused ultrasonic solid-liquid extraction; GPC: gel permeation chromatography; HPLC: high performance liquid chromatography; HRMS: high resolution mass spectrometry; HRPS: high resolution product scan; LDTD: laser diose thermal desorption; LC: liquid chromatography; MAE: microwave assisted extraction; MeOH: methanol; MS: mass spectrometry; MS/MS: tandem mass spectrometry; PLE: pressurized liquid extraction; PSA: primary secondary amine; PuLE: pulverised liquid extraction; QuEChERs: Quick, easy, cheap, effective, rugged and safe; RRLC: rapid resolution liquid chromatography; UHPLC: ultra-high performance liquid chromatography; USE: ultrasound assisted extraction; w.w.: wet weight.

## Data Availability

The study did not report any data.

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
