# Peer review of "An Overview of Analytical Methods to Determine Pharmaceutical Active Compounds in Aquatic Organisms"

_molecules, 2022, doi:10.3390/molecules27217569_

Round 1

Reviewer 1 Report

in figure 1, the authors should give more information on how they obtained such information. Did they search scientific databases? the sole figure do not give relevant information

All scientific names must be written in italics (line 120)

Table 1 is extremely extensive and quite hard to swallow; authors should try to cut the different tables (maybe my biological group) and discuss what is the difference between the methods used.

In section 3 authors should discuss, (even briefly) the difference between the methods described. the latter would give the readers a better overview of which method can suit better their needs

Author Response

In figure 1, the authors should give more information on how they obtained such information. Did they search scientific databases? The sole figure does not give relevant information.

The reviewer is right, and we appreciate this comment. Figure 1 has been elaborated from the literature consulted and compiled in Table 1, to highlight which aquatic organisms have been most and least investigated the determination of PhACs, which we consider could be of interest in view of future research. For clarity, it has been properly specified in the text. Page 2: Figure 1 displays the number of studies devoted to the analysis of PhACs for each group of marine organisms according to the literature consulted in scientific databases. It shows that fishes have been by far the most investigated in this field.”

All scientific names must be written in italics (line 120)

Thank you for the remark. The typo has been corrected: Daphnia magna, Cepopeda, Caldocera and Rotifers”.

Table 1 is extremely extensive and quite hard to swallow; authors should try to cut the different tables (maybe my biological group) and discuss what is the difference between the methods used.

Thank you for your comment. It is true that the table 1 is very extensive due to all the amount and relevant information it contains. Therefore, taking into account your suggestion, the table has been divided according to the biological groups, from “a” to “i”. With this, we have eliminated one column and have achieved shorter tables.

In section 3 authors should discuss, (even briefly) the difference between the methods described. The latter would give the readers a better overview of which method can suit better their needs.

The reviewer is right. Each technique of analysis and sample treatment has its advantages and disadvantages, and although the operative differences between them are described for each specific technique in body text and in the tables, following your recommendation, we have added a brief paragraph at the end of the section where, the most commonly used techniques for PhACs analysis in biota are highlighted to make it clearer for reader. Page 39: “Of all the extraction techniques described in this section, those based on the use of ultrasound (USE and FUSLE) have been the most attractive alternative for the analysis of PhACs in biota (36% of the studies), followed by PLE (30% of the consulted studies). Both techniques are simple, provide automatization, short extraction times and low solvent consumption. For clean-up, SPE using Oasis HLB cartridges has shown to be an efficient method and the most popular used as a clean-up procedure (71% of the studies), regardless of the aquatic organism under study.”

Reviewer 2 Report

The manuscript attempts to capture the main contributions reported by different journals to a special domain of perturbation of life quality in aquatic environment.  From this point of view the work submitted by the authors is useful for researchers involved in this domain, and not only for them.  There are a few observations related to the content of this review, which can be improved for better importance of this topic.  Even if the aim of this version, as the authors emphasized in Abstract, is to offer a summary of the proposed and developed analytical methodologies reported by the literature for the determination of pharmaceutical compounds in aquatic organism, the paper should be more detailed and give a more critical viewpoint.  For a more unitary view of the main analytical methodologies used for the aim of this review some modifications may give a more systematic debate of this topic.  For example:

-  Dispersive solid phase extraction should be discussed as a single section, which has different purposes including cell disruption.

-  Solid phase extraction is however an extraction technique which has various purposes, including clean-up and purification.  Therefore, its broad utilization should be discussed as a single section in 3.3.  The same observation is for QuEChERs.  Overall, the entire section 3.4. should be included in restructured 3.3 with more critical discussion of the various procedures reported by the literature.

-  Derivatization is not included in the content of this review, which can give a more complete view of this domain.

-  Section 4.1. should be more extensive, including discussion on the various retention mechanisms used for sample separation resulted after their preparation.  For example, Hilic is not even mentioned as an alternative for the separation of polar compounds;  ion-pairing mechanism is also missing.  This section should be entirely reformulated as is the core of the analytical methodologies used for the determination of pharmaceutical compounds in biological matrices, which are well structured in the first part of this review.

Author Response

The manuscript attempts to capture the main contributions reported by different journals to a special domain of perturbation of life quality in aquatic environment. From this point of view the work submitted by the authors is useful for researchers involved in this domain, and not only for them. There are a few observations related to the content of this review, which can be improved for better importance of this topic. Even if the aim of this version, as the authors emphasized in Abstract, is to offer a summary of the proposed and developed analytical methodologies reported by the literature for the determination of pharmaceutical compounds in aquatic organism, the paper should be more detailed and give a more critical viewpoint. For a more unitary view of the main analytical methodologies used for the aim of this review some modifications may give a more systematic debate of this topic. For example:

We would like to thank the reviewer for his/her positive appreciations, in particular in consider the topic of interest and useful for researchers both in the field and in other areas.

-Dispersive solid phase extraction should be discussed as a single section, which has different purposes including cell disruption.

The reviewer is right. Points 3.3 and 3.4 have been rearranged for clarity for the reader.

 Solid phase extraction is however an extraction technique which has various purposes, including clean-up and purification. Therefore, its broad utilization should be discussed as a single section in 3.3. The same observation is for QuEChERs. Overall, the entire section 3.4. should be included in restructured 3.3 with more critical discussion of the various procedures reported by the literature.

Thank you for your comment. The reviewer is right. We have changed the title 3.3 to “Sample treatment (extraction and/or clean-up) and restructured it by integrating all concepts without differentiating between extraction and cleaning.

- Derivatization is not included in the content of this review, which can give a more complete view of this domain.

 Thank you for your comment. It is true that derivatization has not been included in this review because 100% of the articles consulted used liquid chromatography due the physicochemical properties of the compounds, in which this step is not necessary.

-Section 4.1. Should be more extensive, including discussion on the various retention mechanisms used for sample separation resulted after their preparation. For example, Hilic is not even mentioned as an alternative for the separation of polar compounds; ion-pairing mechanism is also missing. This section should be entirely reformulated as is the core of the analytical methodologies used for the determination of pharmaceutical compounds in biological matrices, which are well structured in the first part of this review.

 The reviewer is right. We appreciate your suggestion and we have modified the section 4.1 in agreement with your observation, including a discussion of the retention mechanisms employed for separation of the PhACs in the consulted articles and structuring the section. See page 39-40.

Reviewer 3 Report

The manuscript “An Overview of Analytical Methods to Determine Pharmaceutical Active Compounds in Aquatic Organisms” brings an overview of the recent methods for the determination of several groups of PhACs in aquatic organisms. The authors focused on last 10 years and summarize the sample preparation process, both extraction and purification, and instrumental analysis, and pointed out the most widely used, and their advantages and disadvantages. There are a few comments to be answered before acceptance:

1.     The authors should review Table 1., special columns Extraction technique, Clean-up and Analysis. It’s not clear what is in which columns.

2.     In section 3 (line 170) the samples were stored frozen or deep-frozen until analysis. How long the samples were frozen or deep-frozen before being analyzed, whether it was hours, days, or months?

3.     Extraction and preconcentration steps are highly demanded prior to instrumental analysis to increase PhACs concentrations up to values so they can be quantified by analytical equipment and to remove major interferences. In comparison with the ones listed methods, aqueous biphasic systems (ABSs) are a very promising sample pretreatment method, so authors should consider putting some words about them in this manuscript.

4.     Line 454: Specifically, which pharmaceutical was analyzed the most based on the selected literature in this review?

Author Response

The manuscript “An Overview of Analytical Methods to Determine Pharmaceutical Active Compounds in Aquatic Organisms” brings an overview of the recent methods for the determination of several groups of PhACs in aquatic organisms. The authors focused on last 10 years and summarize the sample preparation process, both extraction and purification, and instrumental analysis, and pointed out the most widely used, and their advantages and disadvantages. There are a few comments to be answered before acceptance:

We would like to thank the reviewer for his/her valuable comments. We went throughout the manuscript and made all needed corrections thus improving the overall quality of the manuscript.

1. The authors should review Table 1., special columns Extraction technique, Clean-up and Analysis. It’s not clear what is in which columns.

Thanks for the suggestion. In agreement with the Reviewer 1’s observation, we have modified the tables to better differentiate to make them easier to read.

2. In section 3 (line 170) the samples were stored frozen or deep-frozen until analysis. How long the samples were frozen or deep-frozen before being analyzed, whether it was hours, days, or months?

Thank you for your comment. We consider that it would be of great interest and very useful for the articles to specify this, but after reviewing them to include the information in the review, we found that most of the articles included in the present review did not indicate the time stored until the analysis, so we have not been able to add this information.

3. Extraction and preconcentration steps are highly demanded prior to instrumental analysis to increase PhACs concentrations up to values so they can be quantified by analytical equipment and to remove major interferences. In comparison with the ones listed methods, aqueous biphasic systems (ABSs) are a very promising sample pretreatment method, so authors should consider putting some words about them in this manuscript.

The reviewer is right, and the proposal is very interesting. ABSs have not been listed with the rest of the sample treatment methods because it has not been used in the works consulted. But we have taken your valuable suggestion into consideration, and therefore, this extraction technique has been included as a future perspective.

Page 42: “Although other green techniques should be explored for the extraction of these compounds to further reduce solvent and extraction time, such as aqueous two-phase systems (ABS), which remove volatile organic compounds and have very promising prospects”.

4. Line 454: Specifically, which pharmaceutical was analyzed the most based on the selected literature in this review?

Thank you for your comment, we consider it of great interest to add the information you suggest. We have added the most analyzed drugs from each family. Page 41: Among the PhACs most investigated were antibiotics (ciprofloxacin, trimethoprim and sulfamethoxazole), non-steroidal anti-inflammatory drugs (NSAIDs), analgesics (diclofenac, ibuprofen, naproxen and acetaminophen), antidepressants (venlafaxine), and antihypertensive drugs (propranolol and metoprolol) in this order which also correspond with the most accepted and consumed by the human population”.